# Meiosis-specific distal cohesion site decoupled from the kinetochore

Bo Pan[1], Melania Bruno [2], Todd S. Macfarlan [2] & Takashi Akera [1]✉

Primary constriction of the M-phase chromosome serves as a marker for the kinetochore position. Underlying this observation is the concept that the kinetochore is spatially linked with the pericentromere where sister-chromatids are cohered. Here, we find an unconventional chromatid-cohesion pattern in *Peromyscus* oocytes, with sister chromatids cohered at a chromosome end, spatially separated from the kinetochore. This distal locus enriches cohesin protectors specifically during meiosis, and chromosomes with this additional cohesion site exhibit enhanced cohesin protection at anaphase I compared to those without it, implying an adaptive evolution to ensure cohesion during meiosis. The distal locus corresponds to an additional centromeric satellite block, located far from the satellite block building the kinetochore. Analyses on three *Peromyscus* species reveal that the internal satellite consistently assembles the kinetochore in mitosis and meiosis, whereas the distal satellite selectively enriches cohesin protectors in meiosis to promote cohesion. Our study demonstrates that cohesion regulation is flexible, controlling chromosome segregation in a cell-type dependent manner.

Accurate chromosome segregation during mitosis and meiosis is crucial for maintaining genomic stability and ensuring the faithful inheritance of genetic material across generations. There are at least two fundamental and evolutionarily conserved features of M-phase chromosomes to ensure faithful segregation: (1) the assembly of the kinetochore to interact with the spindle apparatus and (2) the cohesion of sister chromatids to ensure bi-orientation of the chromosome[1–5]. The kinetochore position and the sister-chromatid cohesion site are spatially linked and located on centromeric satellite DNA in many species. Indeed, the primary constriction site, where the sisters are most tightly cohered, is a classic indicator for the centromere/kinetochore position to determine the karyotype of each species, first described by Walter Flemming in 1882[6–8]. At a molecular level, chromosome cohesion is mediated by the cohesin complex, which initially loads along the chromosome axis. Upon mitotic entry, cohesin on the chromosome arm is removed by the prophase pathway, whereas pericentromeric cohesin is protected until anaphase onset where the Separase-mediated cleavage takes place, allowing chromosome

segregation. Kinetochores play a role in this cohesin protection at the pericentromere by recruiting Chromosomal Passenger Complex (CPC) and the Shugoshin (SGO)-PP2A complex to the pericentromere[9–12]. On the other hand, CPC at the pericentromere facilitates kinetochore assembly at the centromere in several organisms[13,14]. Therefore, there are multiple molecular links between the kinetochore and the pericentromere to ensure proper chromosome segregation.

Despite the essential and conserved role of centromeres in chromosome segregation, it paradoxically represents the most rapidly evolving part of the genome[15–18]. The functional consequences of rapid centromere evolution are largely unknown. Particularly, the impact of centromere evolution on pericentromere specification and functions have not been investigated at a molecular level. The *Peromyscus* mouse is an ideal system to tackle this question because of their rapid centromere evolution in both size and position, driving karyotypic diversity across the *Peromyscus* genus[19–21]. *Peromyscus* satellite (PMsat) are satellite repeats that locate at the (peri)centromere region in various *Peromyscus* mouse species[21,22]. Previous studies revealed that

[1]Cell and Developmental Biology Center, National Heart, Lung, and Blood Institute, National Institutes of Health, Bethesda, MD, USA. [2]The Eunice Kennedy Shriver National Institute of Child Health and Human Development, National Institutes of Health, Bethesda, MD, USA. ✉e-mail: takashi.akera@nih.gov

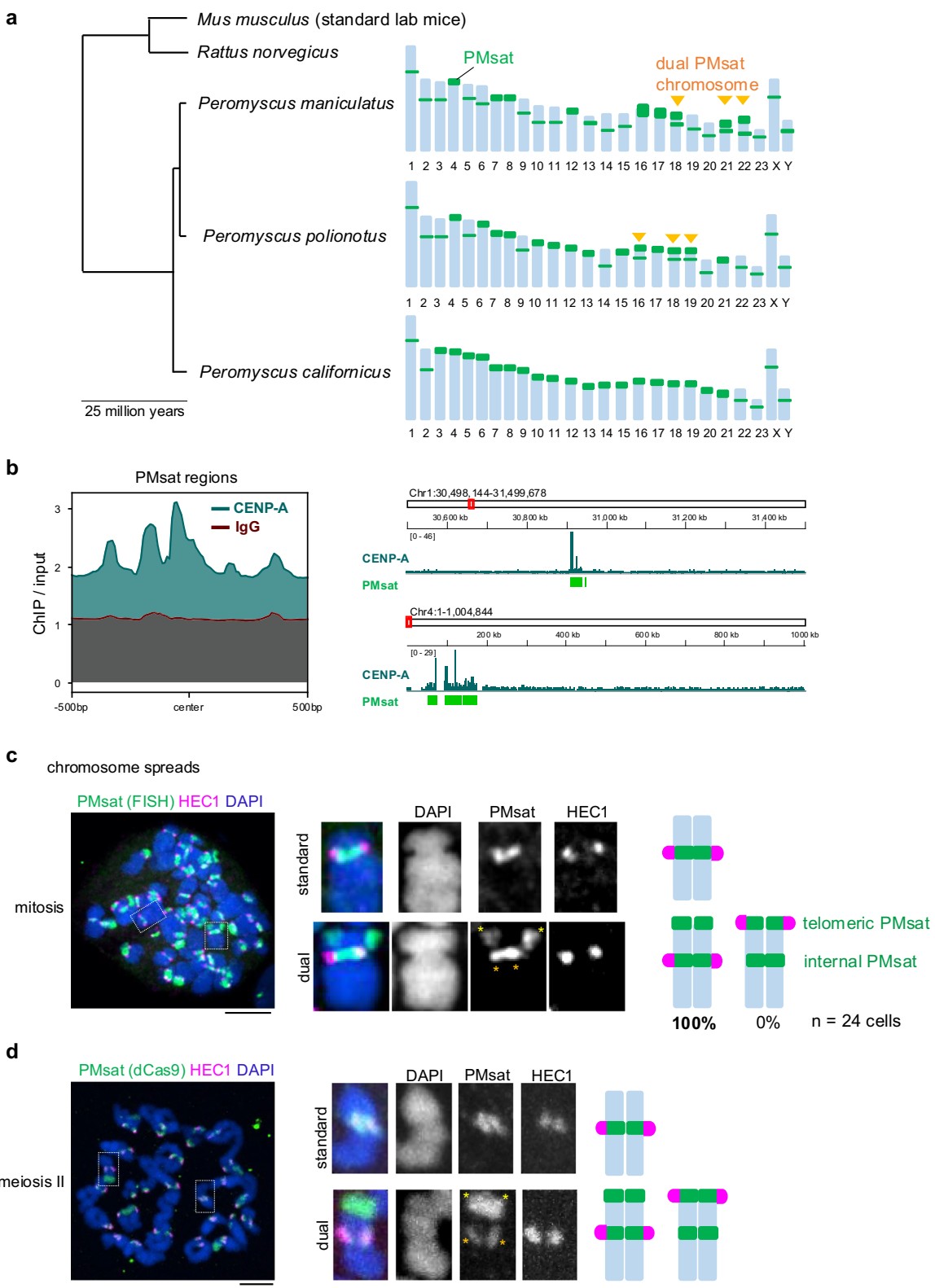

PMsat is present at the (peri)centromeric region of all chromosomes. Interestingly, PMsat is located also at non-centromeric regions proximal to telomeres (hereafter telomeric PMsat) in several chromosomes (e.g., chromosome 18, 21, and 22 in *Peromyscus maniculatus*, hereafter referred to as dual PMsat chromosomes) (Fig. 1a). We took advantage of these naturally occurring chromosomes harboring two blocks of centromeric satellites to investigate the impact of centromere evolution on (peri)centromere specification.

## Results

### PMsat is the centromeric satellite of *Peromyscus maniculatus*

We first confirmed that PMsat is the centromeric satellite for *Peromyscus* mice. Centromeres enrich specialized nucleosomes containing the histone H3 variant, CENP-A, which defines the kinetochore assembly site[23–26]. We enriched CENP-A chromatin from *Peromyscus maniculatus* granulosa cells by chromatin immunoprecipitation (ChIP). High-throughput sequencing and analysis revealed that CENP-A is enriched

**Fig. 1 | Kinetochores assemble exclusively at internal PMsat on dual PMsat chromosomes. a** Phylogenetic tree of mouse species in the *Mus*, *Rattus*, and *Peromyscus* genus. For each *Peromyscus* species, the chromosomal distribution of PMsat is shown based on Smalec et al. [21]. *Peromyscus maniculatus* and *Peromyscus polionotus* but not *Peromyscus californicus* carry chromosomes with two PMsat blocks. **b** CENP-A enrichment at PMsat regions. CENP-A and IgG enrichment on PMsat sequences is provided as ratio of ChIP signal over the input (left). IGV snapshots of CENP-A enrichment (ratio over input) at PMsat regions on two chromosomes (right). **c**, Metaphase chromosome spread using *P. maniculatus* mitotic cells (ovarian granulosa cells) were stained for PMsat (Oligopaint) and a

kinetochore marker, HEC1. **d** *P. maniculatus* meiosis II oocytes expressing dCas9-EGFP and gRNA targeting PMsat were used for chromosome spread and stained for HEC1. The proportion of chromosomes that assemble kinetochores at internal PMsat and telomeric PMsat was quantified; *n* = 24 and 56 cells from at least three independent experiments were examined for mitosis (**c**) and meiosis II (**d**), respectively. The images are maximum projections showing all the chromosomes (left) and optical sections to show individual chromosomes (right); asterisks denote the chromosomal location of internal PMsat (orange) and telomeric PMsat (yellow) on dual PMsat chromosomes; scale bars, 5 μm.

at regions containing the PMsat sequence (Fig. 1b and Supplementary Fig. 1). We observed a strong association of genomic regions enriched for CENP-A binding and containing the PMsat sequence (Supplementary Fig. 1c, d). Furthermore, de novo motif discovery analysis in the sequences underlying CENP-A peaks confirmed the presence of the PMsat consensus sequence (Supplementary Fig. 1e), demonstrating that PMsat is the primary centromeric satellite in *Peromyscus maniculatus*.

### Internal PMsat builds the kinetochore in mitosis and meiosis

We next tested how the centromere position is specified when a chromosome harbors one or two centromeric satellite blocks. HEC1, a major outer kinetochore component, co-localized with PMsat and CENP-A in all standard chromosomes with a single PMsat locus in mitotic chromosome spreads (Fig. 1c and Supplementary Fig. 2b, standard), confirming that PMsat is indeed the centromeric satellite of this species. Interestingly, kinetochores were always and solely assembled on internal PMsat (instead of telomeric PMsat) on all dual PMsat chromosomes (Fig. 1c and Supplementary Fig. 2b), implying (1) a selective pressure to utilize the internal centromeric satellite to form the kinetochore and (2) a silencing of telomeric PMsat to avoid the formation of dicentric chromosomes. Since somatic cells and oocytes can have distinct regulation of centromeric chromatin[27], the kinetochore position was also analyzed in meiosis I and II oocytes. Similar to mitotic cells, oocytes assembled their kinetochores at internal PMsat, demonstrating stable specification of the centromere position (Fig. 1d and Supplementary Fig. 2a–d). This centromere specification pattern was conserved in another species, *Peromyscus polionotus*, which became evolutionary separated approximately 100,000 years ago[28] and has different chromosomes with dual PMsat blocks (i.e., chromosome 16, 18, and 19) (Fig. 1a and Supplementary Fig. 3a,b). Collectively, these results show that the kinetochore position is stably maintained at the internal centromeric satellite block across different tissues and species.

### Telomeric PMsat is the major cohesion site in oocyte meiosis

Compared to centromere specification, how the pericentromere is specified is less studied mainly due the general assumption that kinetochore and the pericentromeric cohesion site are spatially linked. Chromosomes with a centromere in the mid-way (i.e., metacentric chromosomes) generally show the characteristic X-shape morphology during M-phase because sister chromatids are tied together in the mid-way while the chromosome arms are separated (Supplementary Fig. 2e). On the other hand, telocentric chromosomes with their centromeres at the chromosome end show V-shape morphology because they are cohered at one end of the chromosome where the centromere resides. These observations established the concept that the major cohesion site is spatially linked with the centromere regardless of the centromere position. The unique centromere organization of *Peromyscus* chromosomes prompted us to revisit this dogma and test the impact of dual centromeric satellite blocks on the sister-chromatid cohesion pattern. We examined the cohesion site of dual PMsat chromosomes in whole-mount cells where spindle microtubule pulling forces are present (in contrast to chromosome spreads in Fig. 1c, d) (Fig. 2a). In mitosis, sister chromatids were tightly cohered at internal

PMsat (i.e., a single PMsat peak in the line scan) whereas chromatids were separated at telomeric PMsat (i.e., two separate PMsat peaks) (Fig. 2a, mitosis). Therefore, internal PMsat assembles the kinetochore and also serves as the major cohesion site for dual PMsat chromosomes similar to standard chromosomes. In meiosis II, sister kinetochores were more separated compared to mitosis as previously reported[29,30], particularly for dual PMsat chromosomes (Fig. 2a, meiosis II, Supplementary Fig. 3c). We found that dual PMsat chromosomes were consistently cohered at telomeric PMsat, showing a single PMsat peak in the line scan (Fig. 2a, meiosis II dual, and Supplementary Fig. 2d). These results imply that chromosomes with two centromeric satellite blocks can switch over the major cohesion site to a distal centromeric satellite block during female meiosis while the kinetochore position remains stable.

Homologous chromosomes recombine and become connected by chiasmata in meiosis I. If the recombination occurred between two PMsat blocks, the cohesion at telomeric PMsat could be deleterious to the cell by preventing the separation of homologous chromosomes in anaphase I (Supplementary Fig. 4). We did not find any oocytes with such recombination pattern in both *Peromyscus maniculatus* and *Peromyscus polionotus*, implying a mechanism to prevent recombinations between two PMsat blocks. (Peri)centromeric regions usually have lower recombination rates[31–33], and therefore, a similar mechanism could be at play between two PMsat blocks to avoid meiotic failures.

The cohesin complex mediates chromosome cohesion in mitosis and meiosis[34–36]. The unconventional cohesion pattern observed in *Peromyscus* oocytes raised a possibility that telomeric PMsat facilitates cohesin-mediated chromosome cohesion specifically in meiosis. To test this possibility, we first examined the localization of meiosis-specific cohesin subunit, REC8, in oocytes. In multiple organisms, cohesin localizes along the chromosome axis in meiosis I, followed by the Separase-mediated cleavage in anaphase I except for pericentromeric cohesin. Analogous to protecting cohesin from the prophase pathway in mitosis, the SGO-PP2A complex protects pericentromeric cohesin from Separase at anaphase I[9,34,35]. The remaining cohesin at the pericentromere allows the bi-orientation of sister chromatids in meiosis II much like in mitosis. Consistent with other organisms, REC8 cohesin localized on the chromosome axis in meiosis I and at the pericentromere in meiosis II (Fig. 2b and Supplementary Fig. 5a), suggesting that general principles for meiotic cohesin regulations are conserved in *Peromyscus* mice. When we focused on dual PMsat chromosomes in meiosis II, we found that cohesin remains localized at telomeric PMsat in addition to the pericentromere (Fig. 2b, meiosis II dual, and Supplementary Fig. 5b), consistent with our hypothesis. To directly test if cohesin mediates sister-chromatid cohesion at telomeric PMsat, we acutely degraded REC8 by the Trim-Away method[37,38]. We found that both standard and dual PMsat chromosomes fell apart into single chromatids as evidenced by a single kinetochore on the chromosome in contrast to two kinetochores for sister-chromatid pairs in the control condition (Fig. 2c and Supplementary Fig. 5c). These data suggest that cohesion at telomeric PMsat is mediated by the cohesin complex. Dual PMsat chromosomes that were not completely separated by REC8 Trim-Away were predominantly connected

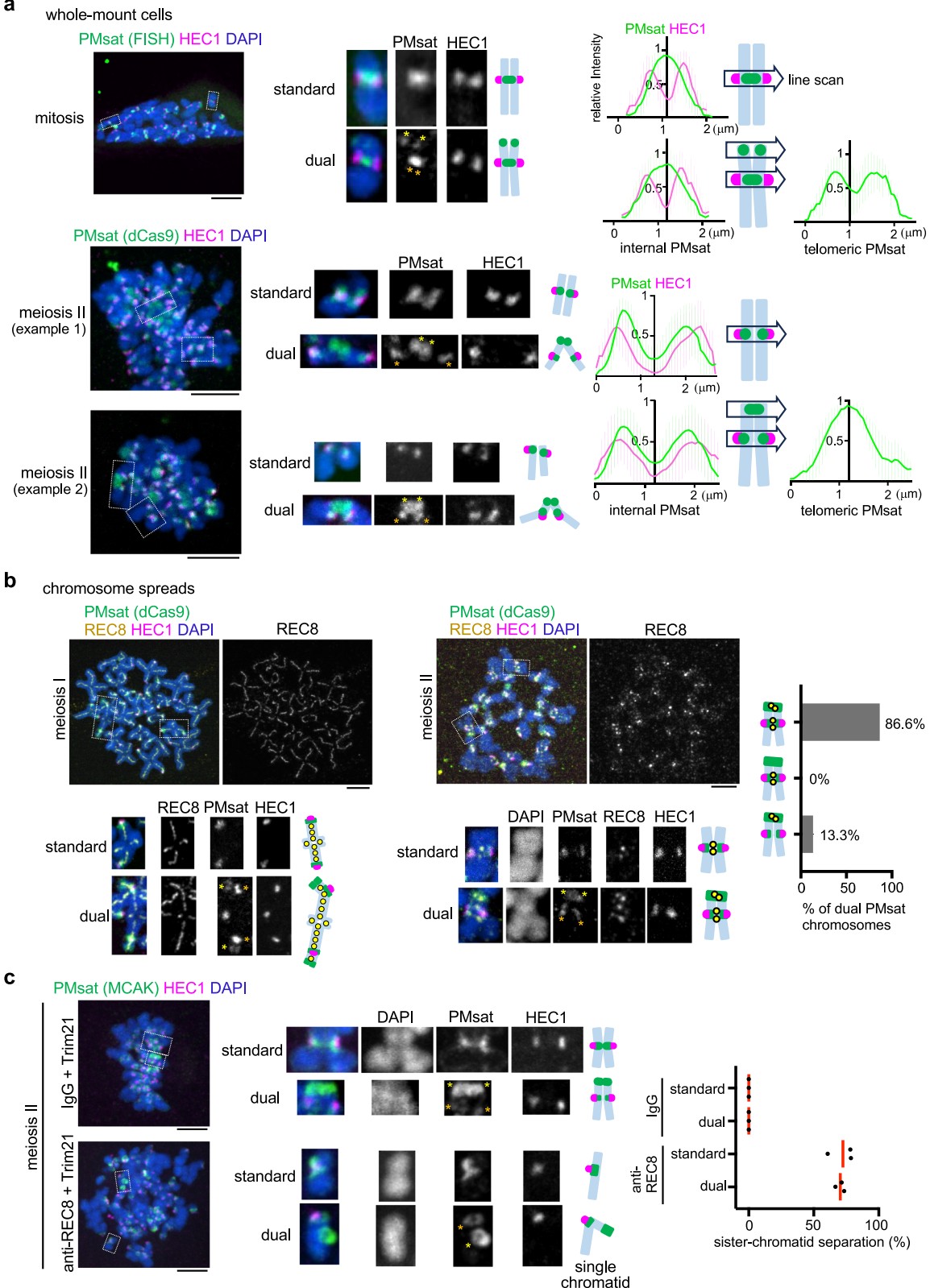

at telomeric PMsat (Supplementary Fig. 5d), supporting the idea that telomeric PMsat is the major cohesion site.

## PP2A-mediated cohesin protection at telomeric PMsat

Cohesin needs to be protected by PP2A activity to maintain its localization at the metaphase I−anaphase I transition[39,40]. We hypothesized that PP2A enriches at telomeric PMsat, in addition to its canonical localization at the pericentromere, to protect cohesin at telomeric PMsat. Indeed, we found that PP2A localized at telomeric PMsat in meiosis I oocytes (Fig. 3a). PP2A levels were slightly but significantly higher at telomeric PMsat compared to internal PMsat, consistent with the observation that telomeric PMsat serves as the major cohesion site of dual PMsat chromosomes in the following meiosis II division. If telomeric PMsat is the major cohesion site

**Fig. 2 | Telomeric PMsat acts as the primary cohesion site in oocytes. a** *P. maniculatus* mitotic cells (ovarian granulosa cells) and meiosis II oocytes expressing dCas9-mCherry with gRNA targeting PMsat were fixed and stained for HEC1. For the mitotic cells, PMsat was labeled by Oligopaint. Line scans of the signal intensities of HEC1 and PMsat across the PMsat loci were performed; *n* = 13 and 46 cells from three and 11 independent experiments were analyzed for mitosis and meiosis II, respectively; lines represent the mean intensities; error bars, SD.
**b** Chromosome spreads using *P. maniculatus* meiosis I oocytes expressing dCas9-EGFP and meiosis II oocytes expressing dCas9-mCherry together with gRNA targeting PMsat were stained with HEC1 and REC8; *n* = 14 and 9 cells from three independent experiments were analyzed for meiosis I and II, respectively. Additional examples of REC8 staining in Supplementary Fig. 5b. **c** *P. maniculatus* meiosis

I oocytes microinjected with mCherry-Trim21 mRNA together with either control IgG antibody or anti-REC8 antibody were matured to meiosis II and fixed and stained for MCAK (a PMsat marker, see Fig. 4c) and HEC1. Chromosomes with a single kinetochore (HEC1) were scored as single chromatids, and the proportion of chromosomes exhibiting sister chromatid separation was quantified; each dot represents an individual experiment; *n* = 23 and 15 cells from three independent experiments for the IgG and REC8 antibody, respectively; red line, mean. The images are maximum projections showing all the chromosomes (left) and optical sections to show individual chromosomes (right); asterisks denote the chromosomal location of internal PMsat (orange) and telomeric PMsat (yellow) on dual PMsat chromosomes, scale bars, 5 μm. Source data are provided as a Source Data file.

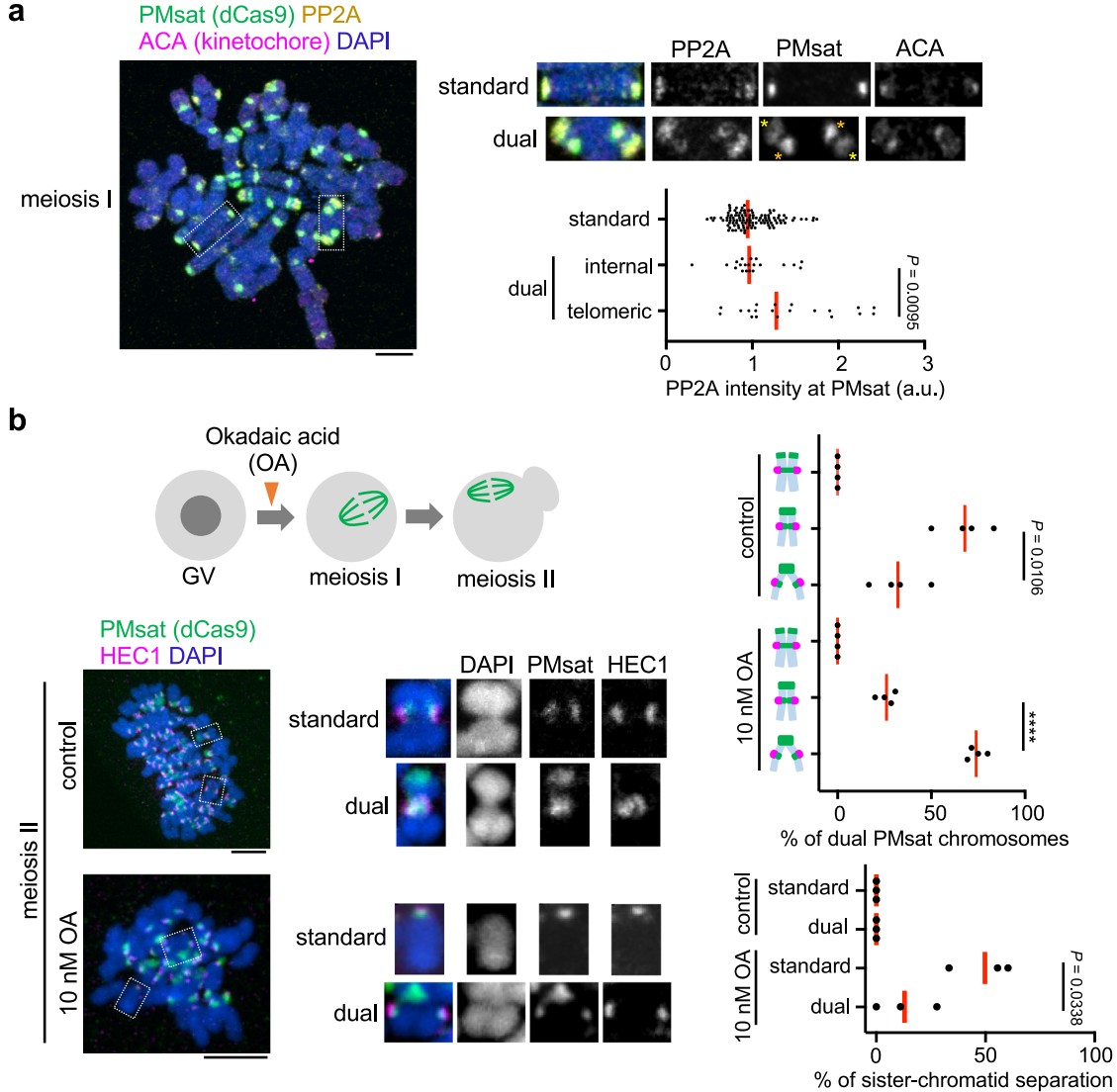

**Fig. 3 | PP2A-mediated cohesin protection at telomeric PMsat. a** Chromosome spreads of *P. maniculatus* meiosis I oocytes expressing dCas9-mCherry with gRNA targeting PMsat were stained for ACA (kinetochore) and PP2A. Signal intensities of PP2A at PMsat were quantified; each dot represents one chromosome; *n* = 152 chromosomes from three independent experiments; unpaired two-tailed Mann-Whitney test was used to analyze statistical significance; red line, median. **b** *P. maniculatus* meiosis I oocytes expressing dCas9-EGFP with gRNA targeting PMsat were treated with 10 nM Okadaic acid (OA), matured to meiosis II, and fixed and stained for HEC1. The proportion of each chromosome configuration of dual PMsat chromosomes (top graph) and sister-chromatid separation (bottom graph; DAPI

and HEC1 signals were used to determine if the chromosome is a single chromatid or sister chromatids) were quantified; each dot represents an individual experiment; *n* = 26 and 33 oocytes from four independent experiments for control and the OA-treated group, respectively; unpaired two-sided t-test was used to analyze statistical significance; red line, mean. Exact *P* values are in the graphs except for *****P* < 0.0001; the images are maximum projections showing all the chromosomes (left) and optical sections to show individual chromosomes (right); asterisks denote the chromosomal location of internal PMsat (orange) and telomeric PMsat (yellow) on dual PMsat chromosomes; scale bars, 5 μm. Source data are provided as a Source Data file.

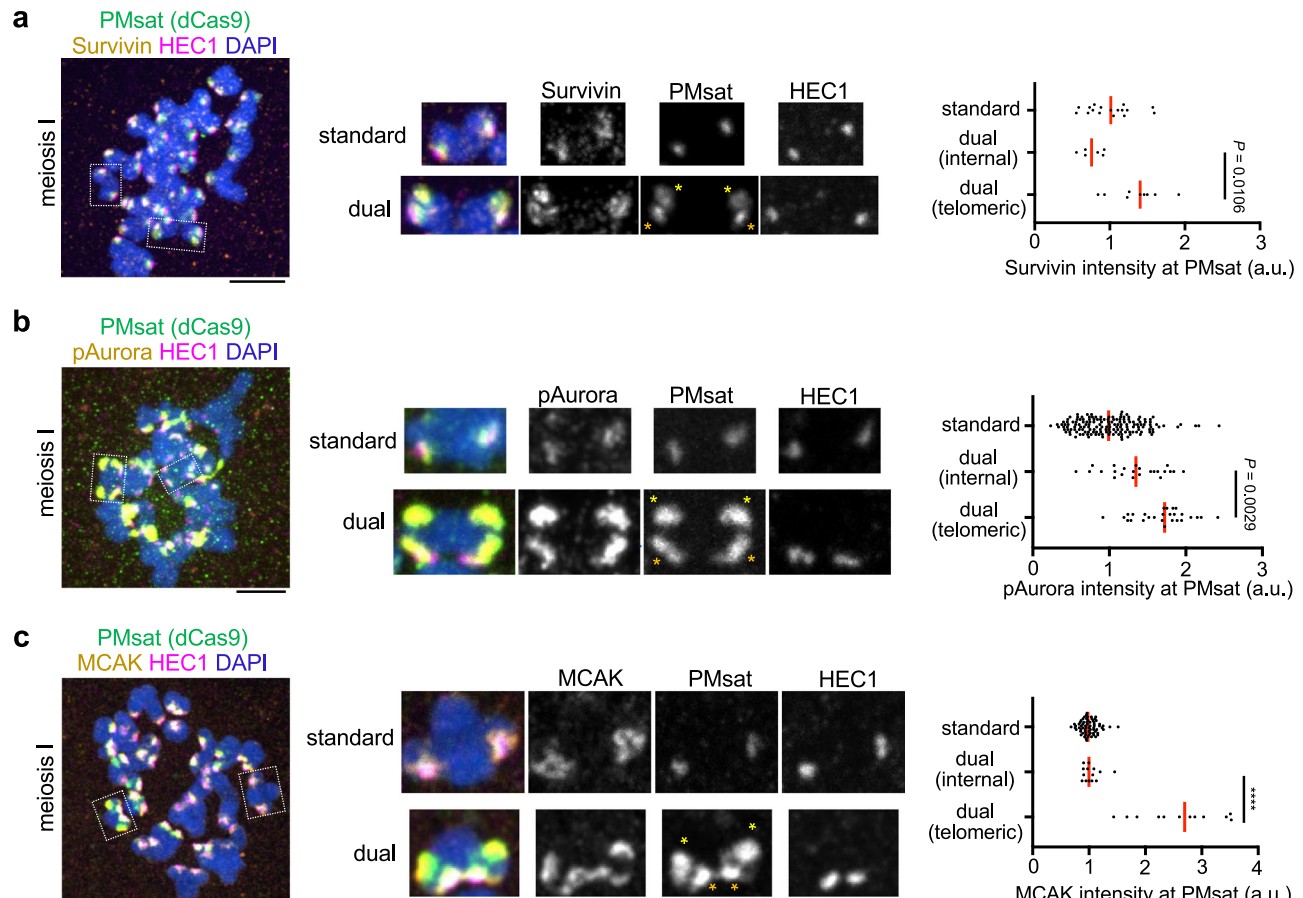

**Fig. 4 | Telomeric PMsat assembles an ectopic pericentromere-like structure decoupled from the kinetochore. a–c** *P. maniculatus* meiosis I oocytes expressing dCas9-EGFP with gRNA targeting PMsat were fixed and stained for HEC1 as well as Survivin (**a**), phosphorylated Aurora kinase (**b**), and MCAK (**c**). Signal intensities of Survivin, pAurora, and MCAK at PMsat were quantified; each dot represents one chromosome; *n* = 32, 210, and 167 chromosomes from three independent experiments were analyzed for Survivin, pAurora, and MCAK, respectively; unpaired two-tailed Mann-Whitney test was used to analyze statistical significance; exact *P* values are in the graphs except for ****P < 0.0001; red line, median. The images are maximum projections showing all the chromosomes (left) and optical sections to show individual chromosomes (right); asterisks denote the chromosomal location of internal PMsat (orange) and telomeric PMsat (yellow) on dual PMsat chromosomes; scale bars, 5 μm. Source data are provided as a Source Data file.

enriching a higher PP2A activity, this locus should be more tolerant to the partial inhibition of the PP2A activity. To test this idea, we treated oocytes with a lower concentration of a PP2A inhibitor, Okadaic acid (OA)[41] (Fig. 3b). Upon partial PP2A inhibition, we observed a substantial increase in the number of dual PMsat sister chromatids only connected at telomeric PMsat (Fig. 3b, top graph). We also noticed that dual PMsat chromosomes are more resistant to the PP2A inhibition compared to standard chromosomes (Fig. 3b, bottom graph). This result implies that carrying an additional block of centromeric satellite could be beneficial for the chromosome to ensure sister-chromatid cohesion during meiosis if cells could prevent the deleterious recombination pattern between two centromeric satellite blocks (Supplementary Fig. 4, see Discussion).

**Telomeric PMsat assembles a pericentromere-like structure**
Given that cohesin and PP2A localized at telomeric PMsat, we wondered if telomeric PMsat also enriches other pericentromeric factors, assembling a pericentromere-like structure that is decoupled from the kinetochore. To test this possibility, we examined the localization of MCAK (mitotic centromere associated kinesin), which is a member of the kinesin-13 family, and the Chromosomal Passenger Complex (CPC) composed of Survivin, Borealin, INCENP, and Aurora B/C kinase[42–44]. Survivin, phosphorylated Aurora (pAurora, labeling active CPC), and MCAK were highly enriched at telomeric PMsat in meiosis I

and II oocytes in addition to their characteristic localization next to the kinetochore (Fig. 4a–c and Supplementary Fig. 6a–c). Similarly, pericentromeric factors localized at telomeric PMsat in *Peromyscus polionotus* oocytes (Supplementary Fig. 7a–c). In contrast, *Peromyscus californicus*, which does not carry dual PMsat chromosomes (Fig. 1a), showed conventional features with the kinetochore and pericentromeric factors always juxtaposed on the chromosome without the formation of additional pericentromere-like structure (Supplementary Fig. 7d). Altogether, these results suggest that the extra centromeric satellite block without kinetochore proteins can recruit pericentromeric factors to establish a pericentromere-like structure, implying a genetic contribution of centromeric satellites to assemble the pericentromere.

**H2A-pT121 recruits pericentromeric factors to telomeric PMsat**
We next asked how telomeric PMsat recruits major pericentromeric factors. While multiple inter-dependencies ensure the enrichment of pericentromeric factors, it is established that two epigenetic marks, histone H3-pT3 and H2A-pT121, are critical for enriching pericentromeric factors in multiple organisms (Fig. 5a)[11,45–47]. Therefore, we tested if these pathways contribute to rewiring the landscape of chromosome cohesion in meiosis. First, we tested the Haspin kinase-mediated H3-pT3 pathway, which recruits CPC through the interaction with the Survivin subunit[46]. We found that inhibiting Haspin by a

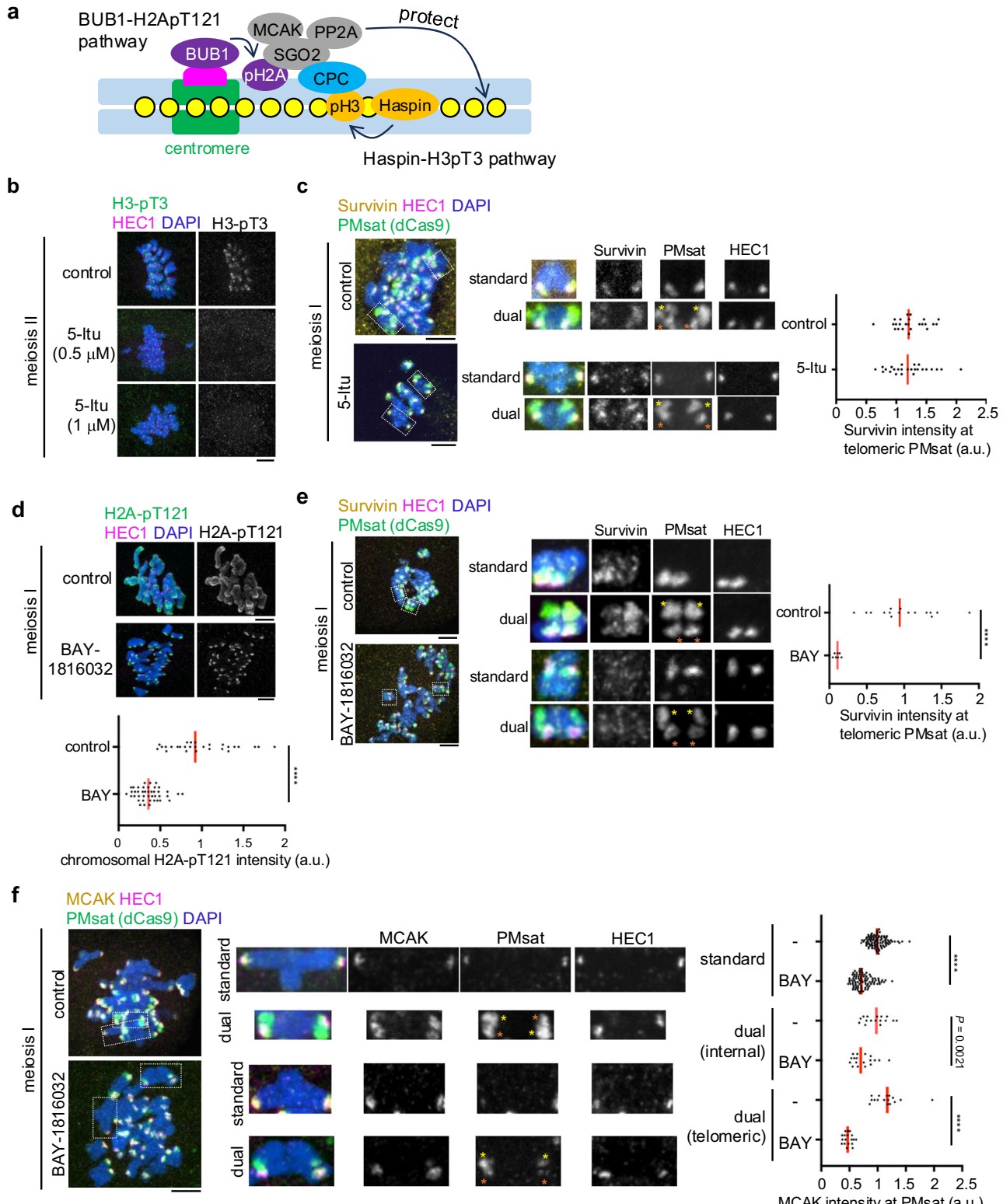

chemical inhibitor, 5-iodotubucidin (5-Itu), abolished H3-pT3 signals on the chromosome (Fig. 5b and Supplementary Fig. 8a)[48]. Furthermore, Haspin inhibition reduced CPC levels on the chromosome arm without significantly impacting its pericentromeric localization (Supplementary Fig. 8b, c), consistent with a previous study using oocytes from lab standard mice, *Mus musculus*[48]. Notably, we did not see a significant reduction in CPC and MCAK levels at telomeric PMsat (Fig. 5c and Supplementary Fig. 8d, e). Therefore, we next tested the

BUB1 kinase-mediated H2A-pT121 pathway, which interacts with SGO2 to recruit MCAK, PP2A, and CPC[11,49,50]. H2A-pT121 signals were detected along the chromosome with a slight enrichment around the pericentromere (Fig. 5d). The chromosomal H2A-pT121 signals were significantly reduced after inhibiting BUB1 by a chemical inhibitor, BAY-1816032 (Fig. 5d)[51]. The BUB1 inhibition reduced CPC levels on the chromosome (Supplementary Fig. 8f), and importantly, reduced CPC and MCAK levels at telomeric PMsat (Fig. 5e, f), suggesting that the

**Fig. 5 | The BUB1 kinase-H2A-pT121 pathway recruits pericentromeric factors to telomeric PMsat. a** Schematic of two pathways recruiting pericentromeric factors. **b** *P. maniculatus* meiosis I oocytes treated with 5-Itu were fixed at meiosis II and stained for HEC1 and H3-pT3. $n$ = 16, 5, and 10 cells from three independent experiments for control, 0.5 μM 5-Itu, and 1μM 5-Itu, respectively. **c** *P. maniculatus* meiosis I oocytes expressing dCas9-EGFP with gRNA targeting PMsat were treated with 5-Itu, fixed at metaphase I, and stained for HEC1 and Survivin. Survivin levels at telomeric PMsat were quantified; each dot represents one chromosome; $n$ = 26 and 23 cells from three independent experiments for control and the 5-Itu-treated group, respectively; red line, median. **d** *P. maniculatus* meiosis I oocytes treated with BAY-1816032 were fixed at metaphase I and stained for H2A-pT121 and HEC1. H2A-pT121 levels on chromosomes were quantified; each dot represents one oocyte; $n$ = 13 and 11 oocytes from three independent experiments for control and the BAY-1816032-treated group, respectively; unpaired two-tailed Mann-Whitney test was used for statistical analysis; red line, median. **e** *P. maniculatus* meiosis I

oocytes expressing dCas9-mCherry with gRNA targeting PMsat were treated with BAY-1816032, fixed at metaphase I, and stained for Survivin and HEC1. Survivin levels at telomeric PMsat were quantified; each dot represents an individual chromosome; $n$ = 16 and 10 oocytes from three independent experiments for control and the BAY-1816032-treated group, respectively; unpaired two-tailed Mann-Whitney test was used for statistical analysis; red line, median. **f** *P. maniculatus* meiosis I oocytes expressing dCas9-mCherry with gRNA targeting PMsat were treated with BAY-1816032, fixed at metaphase I, and stained for MCAK and HEC1. MCAK levels at PMsat were quantified; each dot represents an individual chromosome; $n$ = 122 and 141 chromosomes from four independent experiments for control and the BAY-1816032-treated group, respectively; unpaired two-tailed Mann-Whitney test was used for statistical analysis; red line, median. Orange asterisks, internal PMsat; yellow asterisks, telomeric PMsat; exact $P$ values are in the graphs except for ****$P$ < 0.0001; scale bars, 5 μm. Source data are provided as a Source Data file.

BUB1-H2A-pT121 pathway drives the rewiring of the major cohesion site to telomeric PMsat.

### Formation of ectopic additional cohesion sites is specific to meiosis

While telomeric PMsat serves as the major cohesion sites for dual PMsat chromosomes in female meiosis, sister chromatids appear mainly cohered at internal PMsat in mitosis (Fig. 2a). To understand the mechanisms underlying the difference between mitosis and meiosis, we examined the localization pattern of pericentromeric factors in mitosis, using ovarian granulosa cells. We found that pericentromeric factors and H2A-pT121 were restricted to internal PMsat, localizing between sister kinetochores (Fig. 6a–c), similar to the observations in other model organisms[52,53]. The absence of pericentromeric factors would lead to de-protection of cohesin at telomeric PMsat, explaining why dual PMsat chromosomes are cohered at internal PMsat in mitosis (Fig. 6d). Consistent with this result, we confirmed that pericentromeric factors localize between sister kinetochores and do not form ectopic telomere-proximal pericentromere-like structures in bone marrow mitotic cells (Supplementary Fig. 9).

## Discussion

We have uncovered that centromeric satellites facilitate sister-chromatid cohesion, using *Peromyscus* mice. Because centromeres and kinetochores are usually colocalized, it has been challenging to investigate non-kinetochore roles of centromeric satellites. The unique centromere organization in *Peromyscus* mice allowed us to tackle this question, leading to the identification of the meiosis-specific formation of additional cohesion sites on centromeric satellites that are spatially separated from the kinetochore. The histone H2A-pT121 mark was identified as the main driver to assemble this pericentromere-like structure, recruiting pericentromeric factors such as PP2A, CPC, and MCAK. While it is established that centromeric satellites are not sufficient to establish centromere identity[54], our study implies a previously unappreciated role of centromeric satellites in conferring pericentromere identity. Previous studies on neocentromeres mostly focused on the kinetochore position. Results from this study highlight the importance of revisiting chromosomes with neocentromeres to examine if the original centromeric satellite still enriches pericentromeric factors and contributes to sister-chromatid cohesion in mitosis and meiosis. Indeed, one study has shown that CPC does not fully relocate to the neocentromere from the original centromere in human patient cell lines[55]. Our study revealed a remarkable flexibility in regulating pericentromeric factors in contrast to the stable specification of centromeres and raises two fundamental questions: (1) how is the pericentromere-like structure established in a meiosis-specific manner and (2) Is there an evolutionary advantage to harbor two blocks of centromeric satellites on a single chromosome, and what is it if so?

Our results suggest that H2A-pT121 spreads to the entire chromosome including telomeric PMsat specifically in meiosis to form the additional cohesion site. However, it remains unknown how this epigenetic mark spreads to the entire chromosome in meiosis. BUB1 kinase, which phosphorylates H2A, is restricted to the kinetochore in both mitosis and meiosis (Supplementary Fig. 10) and therefore does not explain the H2A-pT121 spreading. It has been shown that cytoplasmic BUB1 can recruit SGO (the PP2A partner) to chromatin[56]. Therefore, *Peromyscus* oocytes might have overall higher BUB1 activity, which allows BUB1 to act both locally and globally. Phosphorylation levels depend on the balance between the kinase and phosphatase activities. Thus, another possibility is that the activity of the phosphatase that dephosphorylates H2A is relatively weaker in meiosis compared to mitosis. The H2A-pT121 spreading has also been observed in *Mus musculus* oocytes[51]. It would be interesting to explore the biological significance of this drastic change in the epigenetic pattern. In addition, H2A-pT121 is required but not sufficient to explain why pericentromeric factors are restricted to telomeric PMsat. It is likely that there are other factors (e.g., heterochromatin marks) enriched at telomeric PMsat that bridge telomeric PMsat and the formation of the pericentromere-like structure. Future studies would reveal other requirements that are critical to rewire the cohesion landscape in meiosis.

It is interesting to speculate why telomeric PMsat serves as a cohesion site in a meiosis-specific manner. In contrast to mitosis, meiosis undergoes the characteristic two step removal of cohesin from the chromosome. Failure to properly protect cohesin at the pericentromere in anaphase I would result in producing aneuploid gametes and reducing fertility. Furthermore, current and previous studies have shown that sister-chromatid cohesion is in general weaker in meiosis II compared to mitosis[29,30] (Supplementary Fig. 3c). Therefore, one reason to have an additional cohesion site at telomeric PMsat is to ensure sister-chromatid cohesion until meiosis II. Aging reduces cohesin levels on meiotic chromosomes in oocytes, leading to their precocious separation especially for smaller chromosomes[57–60]. We noticed that smaller chromosomes tend to have two centromeric satellite blocks in *Peromyscus* mice (Fig. 1a). Therefore, it is an intriguing possibility that the extra centromeric satellite block serves as a backup mechanism for smaller chromosomes to ensure their sister-chromatid cohesion especially in aged mice. Reinforced cohesion of smaller chromosomes might help to extend the reproductive lifespan of female mice, as it would prevent deleterious mis-segregation of chromosomes during the meiotic divisions. It would be an exciting future avenue to test this idea by removing telomeric PMsat on dual PMsat chromosomes and assess its impact on reproductive lifespan.

Expanded centromeric satellites bias their transmission in animals and plants[16,26,51,61]. Thus, another possibility is that the additional centromeric satellite block increases the selfishness of the

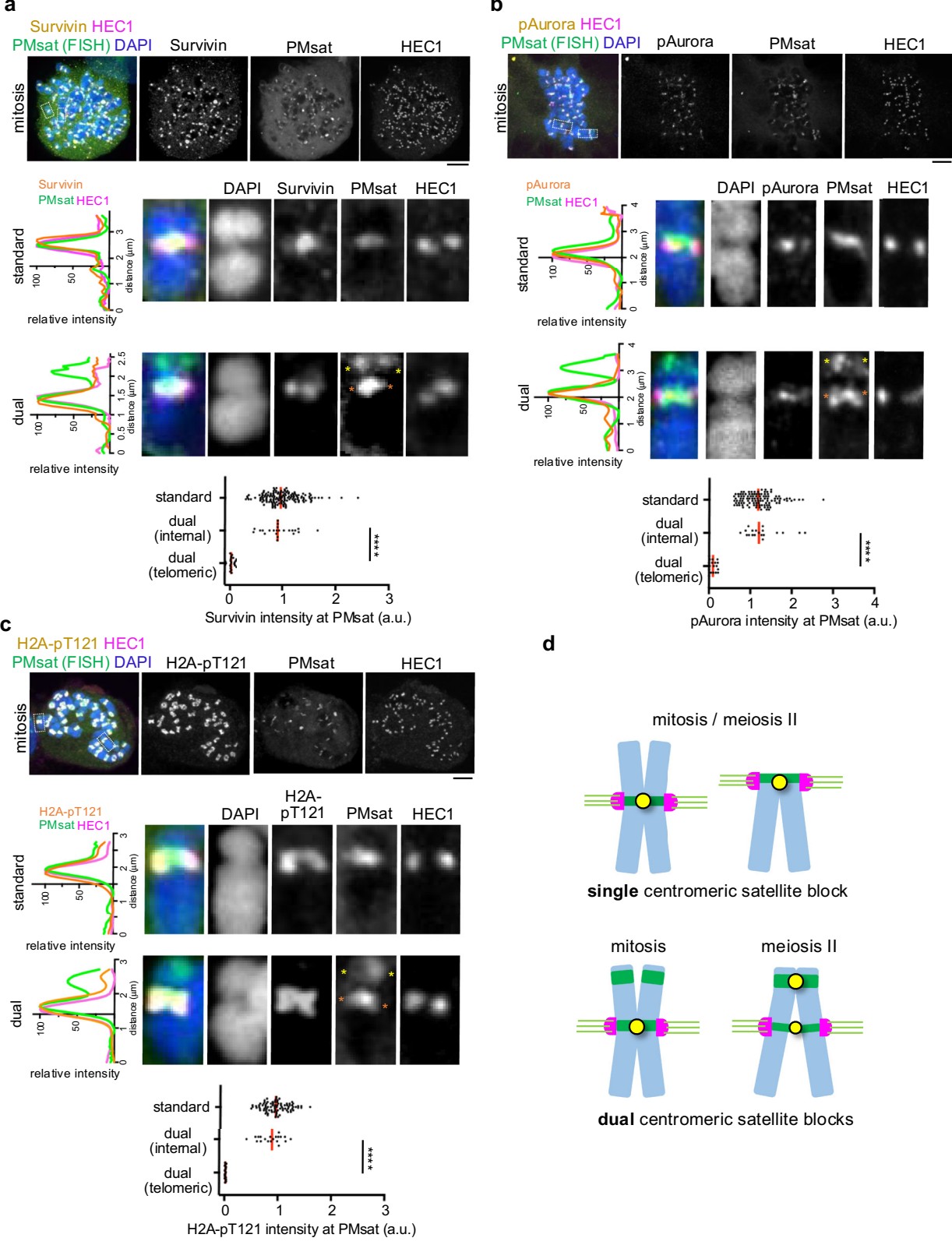

chromosome. MCAK and CPC are the major microtubule-destabilizing activity at the pericentromere that have essential roles to cancel erroneous kinetochore-microtubule attachments[42,44]. This same microtubule-destabilizing activity confers selfishness to mouse centromeres, leading to biased segregation of selfish centromeres[51,62]. Therefore, dual PMsat chromosomes might have been fixed in the population because of their ability to bias their transmission rather than being beneficial to the host.

Altogether, this work provides a conceptual framework to investigate evolutionary forces that shape centromere organization and create karyotypic diversity.

Limitations of the study: an ideal experiment to show the significance of REC8 cohesin at telomeric PMsat would be to specifically degrade REC8 at telomeric PMsat by the REC8-TEV protease system previously established in lab standard mice[63]. However, such experiment requires generating transgenic *Peromyscus* mice, which is

**Fig. 6 | The formation of ectopic additional cohesion sites is specific to meiosis.** **a–c** *P. maniculatus* cells (granulosa cells) arrested in mitosis by Nocodazole were fixed and stained for HEC1 together with Survivin (**a**), phosphorylated Aurora kinase (**b**), or H2A-pT121 (**c**). Immunostained cells were then labeled for PMsat using the Oligopaint technique. *n* = 13, 21, and 11 cells from three independent experiments were analyzed. Line scans of the signal intensities of Survivin (**a**), pAurora (**b**), or H2A-pT121 (**c**) together with PMsat and HEC1 were performed along the chromosome. Signal intensities of Survivin (**a**), pAurora (**b**), and H2A-pT121 (**c**) at PMsat were quantified; each dot represents one chromosome; *n* = 197, 147, and 128 chromosomes from three independent experiments were analyzed for Survivin, pAurora, and MCAK, respectively; unpaired two-tailed Mann-Whitney test was used to analyze statistical significance, ****$P < 0.0001$; red line, median. The images are maximum projections showing all the chromosomes (top) and optical sections to show individual chromosomes (bottom); asterisks denote the chromosomal location of internal PMsat (orange) and telomeric PMsat (yellow) on dual PMsat chromosomes; scale bars, 5 μm. **d**, Model for the centromere and pericentromere specification in mitosis and meiosis when a chromosome carries single or dual centromere satellite block. Source data are provided as a Source Data file.

technically challenging at the moment[64]. Transgenic *Peromyscus* mice would also allow us to manipulate PMsat and robustly visualize weaker REC8 signals in meiosis II[65] (Fig. 2b). It remains unknown why internal PMsat has a slightly but significantly weaker cohesion compared to centromeres of standard chromosomes despite the enrichment of similar PP2A levels (Fig. 3a and Supplementary Fig. 3c). We speculate that other cohesin regulators such as I2PP2A/SET, which inhibits PP2A activity[66], could be differentially regulated between dual PMsat and standard chromosomes.

# Methods

## Mouse strains
*Peromyscus maniculatus bairdii* (BW strain), *Peromyscus polionotus subgriseus* (PO strain) and *Peromyscus californicus insignis* (IS strain) mice were obtained from the *Peromyscus* Genetic Stock Center at the University of South Carolina (https://sc.edu/study/colleges_schools/pharmacy/centers/peromyscus_genetic_stock_center/). Mice were housed in an animal facility with the light/dark cycle of 12 h each and at room temperature with minimal disturbance with a range of 30-70% humidity depending on the season. Mice were euthanized with $CO_2$ followed by cervical dislocation prior to dissection of ovaries. All animal experiments were approved by the Animal Care and Use Committee (National Institutes of Health Animal Study Proposal#: H-0327) and were consistent with the National Institutes of Health guidelines.

## Somatic cell isolation and culture
Ovarian granulosa cells and bone marrow cells were used in this study to examine mitosis. Granulosa cells were used in most experiments because of their ability to proliferate robustly after isolation.

The procedure for isolating and culturing ovarian granulosa cells has been described previously[67]. Briefly, after euthanizing the mice, their ovaries were collected and rinsed three times with M2 media (Sigma-Aldrich, cat# M7167) to remove any adherent fat tissue. The ovaries were then mechanically disrupted to release oocytes and granulosa cells. Following the collection of oocytes, the remaining granulosa cells were collected into a 15 ml tube and allowed to settle at the bottom for 5–10 min. The supernatant was discarded to remove blood cells, and the granulosa cells were then centrifuged at 500 × g for 5 min. The cells were washed extensively with DMEM high glucose GlutaMAX media (Gibco, cat# 10566-016) supplemented with 1× Antibiotic-Antimycotic (Gibco, cat# 15240062). Cells were dispersed by pipetting, washed for two additional times, and then seeded at a density of $0.6 \times 10^6$ cells/ml in 6-well tissue culture-treated plates (Corning, cat# 353046) with DMEM supplemented with 10% FBS (Gibco, cat# A3160501) and 1x Antibiotic-Antimycotic. Cells were cultured in a humidified atmosphere containing 5% $CO_2$ at 37 °C. After 24 h, the medium was replaced with fresh media of the same type to continue the primary culture for ChIP and immunostaining experiments. For immunostaining experiments, we seeded the cells on glass bottom chamber slides (Lab-Tek, cat# 155411) and enriched mitotic cells by double thymidine block and release. At the second thymidine release, 1 μM nocodazole (Sigma-Aldrich, cat# 487929-10MG-M) was added to the medium and the cells were cultured for 16 h before proceeding to standard whole-mount immunostaining or chromosome spread (see below).

The procedure for isolating bone marrow cells was previously described[68], Briefly, bone marrow cells were collected from the femur by inserting a 26-G syringe needle into the cut end of the marrow cavity. Cells were flushed out into 3 ml of pre-warm DMEM high glucose GlutaMAX media (Gibco, cat# 10566-016) supplemented with 10% FBS (Gibco, cat# 10082147), 1 mM sodium pyruvate (Corning, cat# 25-000-CL), and 1× Antibiotic-Antimycotic solution and incubated for 2 h at 37 °C in a humidified atmosphere of 5% $CO_2$ in air. The cells were pelleted twice at 500 x g for 5 min and resuspended in 1× PBS before proceeding chromosome spread (see below).

## Chromatin extraction and ChIP-seq experiment
Granulosa cells were harvested, resuspended in 1x PBS, counted and fixed in 1% formaldehyde for 10 min at room temperature with gentle mixing. Fixation was quenched with 0.4 M glycine for 5 min at room temperature with gentle mixing. Cells were washed twice with cold 1× PBS and cell pellet was frozen on dry ice and stored at -80 °C.

Cells were lysed in cell lysis buffer (5 mM PIPES pH 8.0, 85 mM KCl, 0.5% NP-40, 1× EDTA-free protease inhibitor cocktail (Roche, cat# 5056489001)) for 10 min on ice and homogenized using type-B dounce homogenizer. Released nuclei were pelleted and lysed in nuclei lysis buffer (50 mM Tris-HCl pH 8.0, 150 mM NaCl, 2 mM EDTA pH 8.0, 1% NP-40, 0.5% Sodium Deoxycholate, 0.1% SDS, 1× EDTA-free protease inhibitor cocktail) to release chromatin. Chromatin was sonicated with a Bioruptor® 300 (Diogenode), nuclear debris were then pelleted at 14000 rpm for 15 min at 4 °C, and supernatant was used for chromatin immunoprecipitation. Sonicated chromatin from 20 million cells was used for each ChIP. Dynabeads™ Protein A magnetic beads (Invitrogen, cat# 10002D) were incubated with either custom guinea pig anti-CENP-A (see below) or guinea pig IgG (SinoBiological, cat# CR4) antibodies and washed in 0.5% BSA in 1× PBS. ChIP was performed overnight on rotation at 4 °C. To generate the CENP-A antibody, mixture of two synthetic antigen peptides, MGPRRKPRTPTRRPASC and CRPSSPTPEPSRRSSHL from *Peromyscus maniculatus* CENP-A N-terminal tail, were conjugated with KLH for immunization into three guinea pigs (LabCorp).

Beads were then washed once with low salt wash buffer (0.1% SDS, 1% Triton X-100, 2 mM EDTA pH 8.0, 20 mM Tris-HCl pH 8.0, 150 mM NaCl), twice with high salt wash buffer (0.1% SDS, 1% Triton X-100, 2 mM EDTA pH 8.0, 20 mM Tris-HCl pH 8.0, 500 mM NaCl), twice with LiCl wash buffer (250 mM LiCl, 1% NP-40, 1% Sodium Deoxycholate, 1 mM EDTA pH 8.0, 10 mM Tris-HCl pH 8.0) and twice with TE buffer (10 mM Tris-HCl pH 8.0, 1 mM EDTA pH 8.0). Beads were incubated overnight at 65 °C in elution buffer (10 mM Tris-HCl pH 8.0, 0.3 M NaCl, 5 mM EDTA pH 8.0, 0.5% SDS) with 0.1 μg/μl RNase A (Thermo Scientific, cat# EN0531). Eluates were transferred to fresh tubes and incubated for 2 h at 55°C with 0.3 μg/μl proteinase K (Roche, cat# 3115852001). For the chromatin input sample, elution buffer was added to an aliquot of sonicated chromatin and the sample was treated similarly to ChIP samples. DNA was finally purified with the DNA Clean & Concentrator-5 kit (Zymo Research, cat# D4004).

DNA libraries for NGS were obtained with the ThruPLEX® DNA-Seq Kit (Takara, cat# R400676) with DNA Single Index Kit -12S Set A (Takara, cat# R400695), following manufacturer instructions. Samples

were sequenced as 100 bp paired-end reads on an Illumina NovaSeq 6000 system.

## CENP-A ChIP-seq analysis

Read quality was assessed by fastQC v0.12.1. Reads were aligned to the genome assembly HU_Pman_2.1.3 (GCF_003704035.1) of *Peromyscus maniculatus bairdii* using the Burrows-Wheeler Alignment (BWA) tool v0.7.17 (bwa aln and bwa sampe commands, default settings)[69]. Sam files were then converted into bam files with SAMtools v1.19[70], while removing eventually unmapped and duplicated reads, and retaining only primary alignments (samtools view -F 0 × 4, 0 × 400, 0 × 100, 0 × 800 -b -h file.sam > file.bam). Bam files were sorted and indexed with SAMtools and converted to bigwig normalized to 1x genome coverage (RPGC normalization) for each sample with deep-Tools v3.5.4a[71] (bamCoverage --bam file.bam -o file.bw -of bigwig --binSize 10 --effectiveGenomeSize 2385634842 --normalizeUsing RPGC --extendReads 200). The effective genome size was calculated using the unique-kmers.py command of the tool khmer v2.1.1 (with -k 200)[72–74]. ChIP bigwigs were further normalized by the input using deepTools (bigwigCompare -b1 ChIP.bw -b2 input.bw -o CENPA_input_ratio.bw -of bigwig --operation ratio –skipZeroOverZero --binSize 10). Peaks were called using MACS2 v2.2.7.1[75] (macs2 callpeak -t ChIP.bam -c input.bam -f BAMPE -g 2385634842). Heatmaps and enrichment profiles were plotted using deepTools.

Sequences underlying CENP-A peaks were extracted with getfasta command from BEDTools v2.31.1[76]. De novo motif finding was performed with Multiple Em for Motif Elicitation (MEME) tool from MEME Suite v5.5.5[77] (with options -mod anr -nmotifs 30 -minw 20 -maxw 50 -objfun classic -revcomp -markov_order 0).

To perform enrichment analysis of CENP-A at genomic regions presenting PMsat sequences, a blastn search was performed for the PMsat consensus in the HU_Pman_2.1.3 reference genome assembly. Alignment regions that overlapped or that were at most 10 bp apart were merged using BEDTools merge command. Local Z-score analysis and permutation test ($n = 1000$) to assess the association between CENP-A enriched regions and PMsat regions were performed with regioneR v4.3.1[78].

## Oocyte collection and maturation

Oocyte collection was performed as described previously[79]. Oocytes were handled using a mouth-operated plastic pipette equipped with pipette tips of 75, 100, or 125 μm diameter (Cooper Surgical, Inc., cat# MXL3-75, MXL3-100, and MXL3-125). For in vitro oocyte culture, nuclear envelope (NE)-intact oocytes from female *Peromyscus* mice were collected in M2 media (Sigma-Aldrich, cat# M7167) supplemented with 5 μM milrinone (Sigma, cat# 475840) to prevent meiotic resumption. The oocytes were washed several times in M16 media (Millipore, cat# M7292) to wash out milrinone and transferred to M16 media covered with paraffin oil (Nacalai, cat# NC1506764) to be incubated at 37 °C in a humidified atmosphere with 5% $CO_2$. Only oocytes that underwent nuclear envelope breakdown (NEBD) within 90 min post-release were used for the experiments. For analyses in meiosis I, oocytes were matured for 3-7 h post-release, and meiosis II analyses were performed at least 12 h post-release. Chemical inhibitors were added to the media upon NEBD; BUB1 inhibitor, BAY-1816032 (MedChem Express, cat# HY-103020), at 10 μM; Haspin inhibitor, 5-iodotubercidin (5-Itu) (Cayman Chemical, cat# 10010375), at 0.5 or 1 μM; PP2A inhibitor, Okadaic acid (Sigma-Aldrich, cat# O9381-25UG), at 10 nM.

## Oocyte microinjection

Nuclear envelope-intact oocytes were microinjected with ~5 pl of cRNAs or antibodies in M2 media containing 5 μM milrinone, using a micromanipulator TransferMan 4r and FemtoJet 4i (Eppendorf). Following the microinjection, oocytes were maintained at prophase I in M16 supplemented with 5 μM milrinone for 2-3 h to allow protein expression. EGFP-BUB1 (*Peromyscus maniculatus* BUB1 with EGFP at the N-terminus) were microinjected at 450 ng/μl. cRNAs were synthesized using the T7 mMessage mMachine Kit (Ambion, cat# AM1340) and purified using the MEGAclear Kit (ThermoFisher, cat# AM1908) following the manufacturer's protocols.

To visualize PMsat using the dCas9 technique, dCas9-EGFP cRNA (dead Cas9 with EGFP at the C-terminus, gift from Michael A. Lampson, 800 ng/μl) or dCas9-mCherry cRNA (dead Cas9 with mCherry at the C-terminus, gift from Michael A. Lampson, 800 ng/μl) was mixed with a cocktail of three sgRNAs that target PMsat sequences (PMsat 80, 5′-TAGATATGCCCCGTTTGTGT-3′; PMsat 223, 5′-TTACACTTAGTTGAGGCAAA-3′; PMsat 310, 5′-TCACGATAAACGTGACAAAT-3′; 150 ng/μl each) for microinjection. sgRNAs target part of PMsat consensus sequence that is conserved between *P. maniculatus* and *P. polionotus*. The sgRNAs were synthesized using GeneArt Precision gRNA Synthesis Kit (Thermo Fisher scientific, cat# A29377).

To Trim-Away REC8, mCherry-Trim21 cRNA (*M. musculus domesticus* Trim21 fused with mCherry at the C-terminus, Addgene cat# 105522) at 800 ng/μl and normal rabbit IgG (Sigma-Aldrich, cat# 12-370) or anti-REC8 antibody at 0.2 mg/ml (Invitrogen, cat# pa5-66964) were co-microinjected at the GV stage[37]. REC8 degradation is specific to meiosis II likely due to the lower expression of TRIM21-mCherry in meiosis I (Fig. 2c and Supplementary Fig. 5c).

## Immunostaining of whole-mount cells and chromosome spreads

For whole-mount oocyte staining, meiosis I and II oocytes were fixed in freshly prepared 2% paraformaldehyde (Electron Microscopy Sciences, cat# 15710) in 1× PBS (Quality Biological, cat# 119-069-101) with 0.1% Triton X-100 (Millipore, cat# TX1568-1) for 20 min at room temperature, permeabilized in 1× PBS with 0.1% Triton X-100 for 15 min at room temperature, placed in the blocking solution (0.3% BSA (Fisher bioreagents, cat# BP1600-100) and 0.01% Tween-20 (Thermo Fisher Scientific, cat# J20605-AP) in 1× PBS) overnight at 4 °C, incubated 2 h with primary antibodies at room temperature, washed three times for 10 min with the blocking solution, incubated 1 h with secondary antibodies at room temperature, washed three times for 10 min in the blocking solution, and mounted on microscope slides with the Antifade Mounting Medium with DAPI (Vector Laboratories, cat# H-1200).

For oocyte chromosome spreads, zona pellucida was removed from oocytes using Acidic Tyrode's Solution (Millipore, cat# MR-004-D), and then the oocytes were transferred back to M2 or M16 media and cultured at 37 °C in a humidified atmosphere with 5% $CO_2$ for 30 min to 1 h to allow oocytes to recover. Subsequently, oocytes were fixed with 1% paraformaldehyde, 0.15% Triton X-100, and 3 mM DTT (Sigma, cat# 43815). After the oocytes burst on the microscope slide, the slides were placed in a closed humidified chamber and incubated overnight at room temperature to allow the chromatin to adhere to the slide. The following day, the slides were air-dried completely and then stored in the freezer until immunostaining (see above).

Chromosome spread and whole-mount immunostaining for granulosa cells and bone marrow cells were performed as described above.

The following primary antibodies were used at the indicated delusions for both oocytes and somatic cells: rabbit anti-mouse REC8 (1:200, gift from Michael A. Lampson), mouse anti-human PP2A C subunit (1:100, EMD Millipore, cat# 05-421-AF488), rabbit anti-human Survivin (1:100, Cell Signaling Technology. cat# 2808), rabbit anti-human phospho-Aurora A (Thr288)/Aurora B (Thr232)/Aurora C (Thr198), pAurora (1:100, Cell Signaling Technology, cat# 2914S), rabbit anti-human MCAK (1:1000, gift from Duane Compton), rabbit anti-histone H3-pT3 (1:100, Active Motif, cat# 39154), sheep polyclonal anti human-BUB1 antibody, SB1.3 (1:50, gift from Stephen Taylor), rabbit anti-histone H2A-pT120 (1:2000, Active motif, cat# 39391), mouse anti-human HEC1 (1:200, Santa Cruz, cat# sc-515550), CREST human autoantibody against centromere, ACA (1:100, Immunovision,

cat# HCT-0100), goat anti-GFP antibody conjugated with Dylight488 (1:100, Rockland, cat# 600-141-215).

Secondary antibodies were Alexa Fluor 488–conjugated donkey anti-rabbit (1:500, Invitrogen, cat# A21206) or donkey anti-goat (1:500, Invitrogen, cat# A11057), Alexa Fluor 568–conjugated goat anti-rabbit (1:500, Invitrogen, cat# A10042), or Alexa Fluor 647–conjugated goat anti-human (1:500, Invitrogen, cat# A21445).

## Oligopaint design and oligopaint FISH of mitotic cells

Oligopaints were designed utilizing a modified version of the Oligo-miner pipeline, as previously described[80]. In brief, PMsat sequences, obtained from NCBI and spanning 340 bp, served as the foundation, and Bowtie2 was employed to identify oligos that uniquely mapped to the PMsat locus, utilizing the --very-sensitive-local alignment parameters. Oligo primers used to label PMsat: 5′-TTGGACTGAAGAG AAGCTCCTG-3′ and 5′-TGGGAACAGACGCGAGTG-3′.

To label PMsat with oligopaint probes, cells were fixed in freshly prepared 2% paraformaldehyde (Thermo Fisher Scientific, cat# 28908) in 1× PBS (Quality Biological, cat# 119-069-101) for 20 min at room temperature. Subsequently, fixed cells underwent washing in a blocking solution (0.3% BSA (Fisher Bioreagents, cat# BP1600-100) and 0.01% Tween (Thermo Fisher Scientific, cat# J20605.AP) in 1× PBS. The cells were permeabilized in 1× PBS with 0.1% Triton X-100 (Sigma-Aldrich, cat# TX1568-1) for 15 min at room temperature before returning to the blocking solution.

After immunostaining in the glass bottom chamber slide (Lab-Tek, cat# 155411) (see the previous section), the cells were then fixed a second time with 2% paraformaldehyde in 1x PBS for 10 min at room temperature. Slides were then washed (in coplin jars) with 1× PBS three times for 5 min at room temperature. Subsequently, a primary oligo-paint mix was added to each chamber well, and the chamber was sealed with parafilm. The primary oligopaint mix was composed of 100 pmol of each oligopaint, 1.5 µl of 25 µM dNTPs (New England BioLabs, cat# N0446S), 1 µl molecular grade $H_2O$, 12.5 µl formamide, 4 µl PVSA (Sigma-Aldrich, cat# 278424), 1 µl RNase A (VWR Life Science, cat# E866-5ML), and 6.25 µl DNA hybridization buffer (4 g Dextran sulfate sodium salt (Sigma-Aldrich, cat# D8906-100G), 40 µl Tween, 4 ml 20× SSC, PVSA up to 10 ml), per reaction. After adding primary oligopaint mix, slides were heated to 85 °C on a metal block for 2.5 min and immediately transferred to a 37 °C humidified incubator for an overnight incubation. The following day, parafilm was removed and the slides were washed (in coplin jars) in 2× SSCT for 15 min at 60 °C, in 2× SSCT for 15 min at room temperature, and in 0.2× SSC for 10 min at room temperature. A secondary oligopaint mix was added to each chamber well, and the chamber was sealed with parafilm. The secondary oligopaint mix was composed of 10 pmol of each secondary oligo (IDT, custom synthesized), 6.25 µl DNA hybridi-zation buffer, 12.5 µl formamide, and $H_2O$ up to 25 µl, per reaction. Slides were then transferred to a 37 °C humidified incubator for 2 h. Subsequently, slides were washed in 2× SSCT for 15 min at 60 °C, in 2× SSCT for 15 min at room temperature, and in 0.2× SSC for 10 min at room temperature. A drop of Prolong Diamond Antifade Mountant with DAPI (Invitrogen, cat# P36966) was added to each chamber well.

## Confocal microscopy and image analysis

Fixed oocytes, bone marrow cells, and granulosa cells were imaged using a Nikon Eclipse Ti microscope. The microscope was equipped with a 100×/1.40 NA oil-immersion objective lens, a CSU-W1 spinning disk confocal scanner by Yokogawa, an ORCA Fusion Digital CMOS camera from Hamamatsu Photonics, and controlled laser lines at 405 nm, 488 nm, 561 nm, and 640 nm via NIS-Elements imaging soft-ware by Nikon. Confocal images were captured as Z-stacks at 0.3 µm intervals, and these images were presented as maximum intensity Z-projections unless specified in the figure legend.

For image analysis, Fiji/ImageJ (NIH) software was employed. First, optical slices containing chromosomes were combined to generate sum intensity Z-projections for subsequent pixel intensity quantifications. Signal intensities on the entire chromosome (Survivin and H2A-pT121) were quantified by creating masking images using the DAPI staining. Signal intensities were integrated over each slice after the background signal subtraction. To specifically quantify centromeric signal intensities (PP2A, Survivin, pAurora, MCAK, BUB1, H2A-pT121, and H3-pT3), ellipses were delineated around PMsat or the kinetochore (based on the HEC1 staining) on each chromosome. Signal intensities were then quantified within each ellipse after the background signal subtraction.

## Statistics and reproducibility

Data points were pooled from three independent experiments in most experiments, and the exact number of independent experiments for each experimental group is listed in Supplementary Table 1. Data analysis was performed using Microsoft Excel and GraphPad Prism 10. Scattered plots and line graphs were created with GraphPad Prism 10. unpaired two-tailed Mann-Whitney test and unpaired two-sided t-test were used for statistical analysis unless specified in the figure legend, and the exact $P$ values are shown in each figure. The sample size was chosen based on current practices in the field. Randomization is built into the experiments because each animal was chosen from a different litter and mating pair and no data was excluded and all cells were imaged at random.

## Reporting summary

Further information on research design is available in the Nature Portfolio Reporting Summary linked to this article.

## Data availability

Raw data files for the DNA sequencing analysis have been deposited in the NCBI Sequence Read Archive (SRA) and are available under project accession number SRA: PRJNA1196496. Other data required to repro-duce the results in the current study are available at Figshare [https://doi.org/10.25444/nhlbi.28001618]. Source data are provided with this paper.

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

## Acknowledgements

We thank Alexander E. Kelly for comments on the manuscript, Rachel J. O'Neil for discussion and PMsat reagents, Duane Compton for the MCAK antibody, Michael A. Lampson for the REC8 antibody and the pIVT-dCas9-EGFP and pIVT-dCas9-mCherry plasmids, Stephen Taylor for the BUB1 antibody, R. Zaak Walton for establishing and maintaining *Peromyscus* mouse colonies, and the Akera lab members for discussion. This work is supported by the Intramural Programs of National Heart, Lung, and Blood Institute (1ZIAHL006249) (T.A.) and the *Eunice Kennedy Shriver* National Institute of Child Health and Human Development (1ZIAHD008933) (T.S.M.) at the National Institutes of Health (NIH).

## Author contributions

Conceptualization, T.A.; Methodology, B.P. and M.B.; Investigation, T.A., B.P., M.B., and T.S.M.; Writing—Original Draft, B.P.; Writing—Review & Editing, T.A., B.P., M.B., and T.S.M.; Funding Acquisition, T.A. and T.S.M.; Resources, T.A. and T.S.M.; Supervision, T.A. and T.S.M.

## Funding

## Competing interests

The authors declare no competing interests.
