## [Transparent Peer Review file · Nature Communications]

Meiosis-specific distal cohesion site decoupled from the kinetochore

Corresponding Author: Dr Takashi Akera

Version 0:

Reviewer comments:

Reviewer #1

(Remarks to the Author)

This is a review of the manuscript entitled "Meiosis-specific decoupling of the pericentromere from the kinetochore" from Takashi Akera's laboratory.

The main finding of this paper is that meiotic sister chromatid cohesion is separable from mitotic kinetochore assembly. In addition to this top-level finding, there are other interesting and important other findings. These include: crossover regulation prevents exchange between the two repeat arrays; meiotic sister cohesion depends on PP2A recruitment at the non-kinetochore array; other pericentromere factors are recruited to the non-kinetochore array; BUB1 drives establishment of the telomeric pericentromere. These are amazing findings. They lead to many more questions, which the authors are likely to investigate in future work. These questions are of major importance to multiple chromosome biology fields and include: How are sites of cohesin retention specified in meiosis I? How is the kinetochore assembly site specified during successive mitotic divisions? How is the kinetochore assembly site "remembered" during the meiotic divisions? How do different organisms "know" where to allow crossing over, and how are crossovers "counted?" Though these are old questions, they are important ones, and the current work provides a new experimental system with which study them.

The paper is well-written and clearly reasoned. There are several points on which the data is suggestive but not ironclad (for example, that cohesion "switches" to a new location), so I advise the authors to constrain their claims as much as possible. Even with simplified claims, the manuscript stands out as an important and exciting contribution. Here, I make suggestions primarily related to presentation. I support publication of this manuscript with minor revisions.

Major questions:

- A main finding is that telomeric PMSats remain cohered in MII due to cohesin retention throughout MI. If the authors were to label another region of the chromosome with dCas9 (histone genes? Other heterochromatin region?) that is not under tension, would this appear to be cohered in MII? Can we be sure the telomeric PMSat is truly cohered and not just "not-under-tension?" Agreed that the separation pattern at the PMSat is different in mitosis and meiosis as shown, but these are different experiments from a technical perspective.

- o Experiments to address this: REC8 TRIM-away (see comments below) and okadaic acid (also see below)

- o Perhaps there is not a perfect experiment short of recruiting an inducible Cas9-TEV, but the authors should address this difficulty in the text.

- There is a disconnect between the language and the images shown in figure 2A. The authors say the major cohesion site switches over to the telomeric PMSat arrays in the dual PMSat chromosomes in meiosis II, which implies that centromeric REC8, though it persists to MII in these chromosomes, is not as cohesive as it is in normal chromosomes. However, the HEC1 dots in Figure 2b are separated to a similar degree in standard and dual chromosomes. This is confusing. REC8 persistence at the telomeric array is very cool, but it's not entirely clear what the function of either pool truly is in meiosis II.

Minor points:

- Line 90: "demonstrating the epigenetic memory across soma and germline" Wouldn't one need to show the telomeric PMSat is competent for CENP-A assembly to know this for sure?

- Line 144: "ectopically" seems to me to be the wrong word, as it implies incorrect or non-natural localization.

- The TRIM-away experiment for REC8 is nice, but it is not described at which stage the injection was done, and the MI phenotype is not shown. One assumes there are already major problems at this stage. There is also no direct confirmation

that REC8 is depleted. This experiment is a nice way of confirming that REC8 holds sisters together in MII (whether at PMSat or centromeres), but it could be tightened up slightly, at least in the way it is described.

- Image analysis: for figure 2A and similar, I am confused about how the authors distinguished cohered vs. non-cohered PMSat regions. For instance, the PMSat regions on the “standard” chromosome in 2A (MII) look quite separated and are counted as together (not quantified), but then in the “dual” examples, similarly separated, and these are counted as separated. At least the methods section should explain these image-based decisions clearly.
- Okadaic acid experiment: this is a nice demonstration that phosphatase activity protects sister cohesion that survives to MII. However, without imaging REC8 protein, it is not conclusive that it is cohesin that holds together the telomeric PMSats. The authors should mention this clearly.
- Figure 5 – the imaging, especially for the 5-ltu experiments, are not totally clear to me. An additional point: can the quantification in 5b and in 5d be placed next to/on top of each other to show off the difference with the two compounds more clearly? This is a key comparison.
- Can the schematic from extended data figure 4c be moved to main? This is such a nice illustration of what is going on.
- Acknowledgments: “Stephen Taylor” misspelled as “Stephan”
- Throughout the manuscript, there are very minor grammatical/punctuation errors that can be corrected in copy editing/proofing.

Note: I do not have experience in oocyte imaging, so I urge the editors to consult with a reviewer or external advisor with experience in this area to fully evaluate methodological details and the quality of the data. For the sake of clarity: these look good to me but should be checked by a true expert in this area.

Reviewer #2

(Remarks to the Author)

Pan et al. investigate unconventional pericentromere specification separate from the kinetochore in *Peromyscus* mouse oocytes. They show that PMSat, satellite repeats found at centromeres in various *Peromyscus* species, serves as the DNA scaffold for kinetochore assembly by enriching CENP-A. In both mitosis and meiosis, although some chromosomes have two PMSat loci, one internal and one telomeric, the kinetochore is assembled only on the internal PMSat. Interestingly, however, telomeric PMSat acts as the major persistent cohesion site between sister chromatids at meiosis II. Telomeric PMSat enriches the cohesin protector PP2A at meiosis I, together with the pericentromeric proteins Aurora (CPC) and MCAK. The recruitment of CPC and MCAK depends on the BUB1-H2A-pT121 pathway.

Overall, this manuscript describes a novel and interesting observation of pericentromere specification spatially separated from the kinetochore. Although the mechanism by which telomeric PMSat can enrich pericentromeric proteins specifically in meiosis is not yet very clear, they provide a molecular pathway required for the enrichment. The discussion of the possibility that telomeric PMSat acts as a backup block for smaller chromosomes to protect sister chromatid cohesion against aging is very interesting. The main claims of the manuscripts are well supported by appropriately designed experiments. The results presented in the manuscript will attract a wide range of readers in the field of chromosome biology and reproductive biology. I would support publication in *Nature Communications* if the authors could address the following comments.

1. The authors claim that telomeric PMSat is the major persistent cohesion site at meiosis II. In support of this, they show that PMSat cohesion is more tolerant to treatment with the PP2A inhibitor Okadaic acid (OA). However, the effect of OA treatment is not specific to cohesion protection and is unlikely to mimic physiological conditions. I wondered if they could test REC8 TRIM-Away to partially degrade cohesin, which could mimic a condition of aged oocytes, and see if the PMSat site helps maintain sister chromatid cohesion. The authors have successfully achieved REC8 TRIM-Away (Fig. 2c), so it seems possible to find a condition where REC8 is partially degraded. If such an experiment is technically difficult, it could be informative to show how unseparated sister chromatids looked like after REC8 TRIM-Away in Fig. 2c - did they have persistent cohesion at telomeric PMSat?

2. Related to the above, can they show quantification of REC8 signals on internal and telomeric PMSat, at meiosis I and II?

Minor points:

1. In figure panels, it is not clear what area of the whole image is magnified in the inset images. The authors could consider putting a rectangle to indicate the magnified area. It would be informative to include a scale bar for the magnified images.

2. Fig. 6a-c. If I understand correctly, these line scan data show single representative chromosomes, although according to the legend the authors have images of 13, 21, 11 cells from three independent experiments. Consider adding a quantification of fluorescence signals measured on internal and telomeric PMSats in all the images, to be consistent with what is done in Fig. 4.

Reviewer #3

(Remarks to the Author)

This manuscript by Pan et al (Meiosis-specific decoupling of the pericentromere from the kinetochore) describes a functional characterization of chromosomes that harbor in addition to the regular pericentromeric regions, similar regions elsewhere in

the same chromosomes, close to the telomeres. These chromosomes were analyzed in mouse species of the genus *Peromyscus*, approximately separated by 25 million years of evolution from the common laboratory mouse *Mus musculus*. It has been known for quite some time that the low-complexity DNA sequence underlying the centromeric regions in many higher eukaryotes is neither sufficient nor absolutely required for establishment of a functional centromere. This has led to the notion that centromeres are determined epigenetically, mainly defined by the presence of the centromeric H3 variant CENP-A. CENP-A deposition results in the recruitment of other proteins, which are part of the inner kinetochore and which in turn serve as an assembly platform for the KMN complex that mediates attachment of microtubules of the spindle apparatus during cell divisions. The extended regions surrounding the core centromere – the pericentromeres – are also characterized by repetitive regions, retroelements, transposons and a low density of expressed genes in these heterochromatic settings. For a functional aspect, pericentromeres are known to be enriched in factors providing sister chromatid cohesion – the cohesin complex. The sequences underlying the pericentromeric regions are also not exclusively found in this area, but information about functional relevance of these sequences elsewhere in the genome is scarce. Here, the authors provide evidence that these regions do not suffice to assemble kinetochores but are enriched in factors required for establishment and maintenance of sister chromatid cohesion. Most interestingly, this does not apply for mitotic divisions, but appear to be specific for meiotic divisions (female meiosis it is; male meiosis was not looked at). The studies were mainly conducted using the mouse species, *Peromyscus maniculatus*. The typical pericentromeric sequence signature (called PMSat), which usually occurs only once around the centromere, was found twice in three out of 25 chromosomes (called dual PMSat chromosomes) in this species. The authors diligently used various approaches to ascertain the presence of bona fide PMSat sequences at regions distal to centromeres (in the following telomere proximal PMSat). These telomere proximal PMSats were thoroughly confirmed by either FISH experiments or by targeting a nuclease dead version of Cas9, fused to EGFP, to PMSat regions. Combined FISH/IF and Autofluorescence/IF experiments revealed that the telomere proximal PMSat sequences do not trigger the formation of kinetochores as evidenced by a lack of staining against the outer kinetochore marker HEC1. However, these regions do mediate cohesion up to the metaphase-anaphase transition, but do this exclusively in female meiosis. This is exemplified by visualizing individual metaphase II chromosomes, which are cohesed at the telomere proximal PMSat but not at the centromere. These type of chromosomes were not observed during mitotic divisions. This cohesion is dependent on cohesin as shown by TRIMaway mediated destruction of the cohesin kleisin subunit REC8 resulting in cohesion loss. The persistence of cohesion at these sites up to metaphase of meiosis II requires cohesin protection from the onslaught of separase during meiosis I. The authors show that protection is due to recruitment of PP2A to telomere proximal PMSat. This recruitment is paralleled by localization of components of the CPC and of the kinesin-13 MCAK to the telomere proximal PMSats. The authors also provide evidence that this ectopic recruitment only occurs in animals carrying PMSats, as it is also observed in chromosomes derived from *P. polionotus*, which also carries three chromosomes with telomere proximal PMSats, but not in chromosomes from *P. californicus*, which does not contain chromosomes with extra regions containing PMSats. To gain insight into factors required for the ectopic localization of typical pericentromeric factors at the ectopic PMSats, the authors inhibited two kinases known to introduce histone phosphorylations involved in recruitment of CPC and MCAK. Notably, inhibition of BUB1, which phosphorylates histone H2A at threonine 121, significantly reduced the CPC component, and (less convincing – see below) MCAK at the telomere proximal PMSats.

I feel that the manuscript thoroughly addresses the relevant questions, the experiments are carried out in a sound way and the conclusions drawn are of significant interest for a broad readership. However, I do have a few issues that need to be addressed before the paper can be published.

1. Pericentromeres are defined as the genomic regions surrounding the centromere (from the Greek word peri meaning around, surrounding). Thus, the occurrence of DNA sequences underlying pericentromeres elsewhere in the genome, far away from centromeres, does not justify calling these regions “pericentromeres”, even if they resemble bona fide pericentromeres in a functional sense. Additional ectopic and functional centromeres are called neocentromeres. Thus, I would strongly suggest that the authors do not call the telomere proximal PMSat containing regions pericentromeres as they do at numerous occasions in the text. Even the title of the paper is misleading in this respect. I would suggest sth like “Meiosis-specific persistence of chromatid cohesion at distal regions containing pericentromere-specific sequence repeats” I acknowledge that this sounds somewhat bulky, but it is less misleading. Also “pericentromere specification” (found e.g. in the abstract, lines 18 and 32) and “rewiring pericentromeres” (lines 180 and 193) should be replaced by an alternative term. This is particularly important, since the present studies have looked mainly at one functional aspect of pericentromere function – cohesion. But there are other functions the authors have not investigated. For example, pericentromeric heterochromatin plays a role in the three dimensional packaging of chromatin within the nucleus – chromocenter formation (e.g. Jagannathan et al (2018) <https://doi.org/10.7554/eLife.34122>)
2. Along those lines... “Colocalization” in IF experiments describes the fact that signals corresponding to the same pixels in two channels of the same R.O.I.s indicate presence of two proteins at the same spot within the cell. The CPC is known to localize to the inner centromere – in fact, the CPC component INCENP has derived its name exactly because of this localization pattern. This is exactly, what the authors show, e.g. in Extended Fig. 8. The signals for HEC1 and MCAK/Survivin/pAurora are distinct. Thus, it is not correct to call the localization pattern of CPC components as “colocalization with the kinetochore” (lines 201, 205 and 703).
3. The specificity of the ectopic cohesion site in female meiosis is intriguing. However, the authors need to point out that they can make this point only for female meiosis, as they have not looked at male meiosis
4. In the introduction (lines 41 ff) the authors dwell on cohesion as being restricted to the centromeric/pericentromeric regions. However, this is only true for cohesion at metaphase; cohesion is established along the entire chromosomes and removed in steps (prophase pathway vs. proteolytic degradation at the metaphase-anaphase transition). The authors fail to mention this.
5. Line 381. The terms “left” and “right” do not correspond to the assembly in Figure 6. Line 382. Asterisks are not present in

Fig. 6.

6. Figure 4. The schematic would be much more appropriate to be part of Figure 5, as the BUB1-H2ApT121 and Haspin-H3pT3 pathways are investigated in Figure 5.

7. Some of the critical data shown in Figure 5 are not represented appropriately in the text or are not convincing. The H2A-pT121 staining in Fig 5c shows a strong accumulation of the signals at the pericentromeres. This is not mentioned in the text ("signals were detected along the chromosome" – line 190). In addition, the reduction of the MCAK signals on dual chromosomes after BUB1 inhibition does not look too convincing, particularly because in the control case (Fig. 5e, upper panel), a distinction between internal and telomeric MCAK signals is almost impossible. I would suggest providing a more appropriate image.

8. The suggestion that the presence of the telomere proximal PMSats may result in a more stable cohesion of small chromosomes during female meiosis (metaphase arrest) is intriguing. The authors support this notion by a significantly extended reproductive lifespan of ectopic PMSats containing *P. maniculatus* and *P. polionotus* individuals when compared to *M. musculus*. This may be, however, a pure coincidence, particularly given the evolutionary distance of 25 million years. It would be more informative to look at the reproductive lifespan of the closely related species *Peromyscus californicus*, which also lacks ectopic PMSats. In addition, even if it is technically very challenging, one could at least mention an experimentally based answer to this question: Introduce ectopic PMSats in the *P. californicus* genome or remove/inactivate these regions in *P. maniculatus* and assess whether this has an influence on the reproductive lifespan.

9. The quantification of PP2A signal intensities on control (standard) chromosomes is identical to the internal pericentromeres on dual chromosomes (Fig. 3A). This should result in similar protection of cohesion against separase activity in meiosis I. How do the authors then explain the frequent occurrence of dual chromosomes, which are separated at the centromeric regions in meiosis II (Fig. 2A)?

10. Extended Fig. 2b. The CENP-A signals of the selected dual chromosome are almost invisible. The assignment then of internal vs. telomeric PMSat appears somewhat arbitrary. If the very weak CENP-A signals are used for this distinction, one could also argue that in this case all PMSat regions exhibit HEC1 signals (4 distinct dots in the anti-HEC1 staining).

11. Extended Fig. 4c. It is an important point that recombination between the centromere and the telomere proximal PMSat is deleterious. The authors have looked at a total of 18 cells, meaning 54 potential candidate chromosomes and they have seen no indication for recombination based on IF images. Given the small size of these chromosomes, it would be hard to detect such a constellation anyways. It would be much more informative to look at molecular markers and determine recombination rates.

Version 1:

Reviewer comments:

Reviewer #1

(Remarks to the Author)

Pan et al. have address most of my comments, including major concerns.

I have the following responses to comments and make minor suggestions for improving the manuscript.

Based on this comment, we revised the method section (Line 411) and also showed that there is no detectable chromosome separation in MI, likely because the TRIM21-mCherry expression is not high enough at this stage (Extended Data Fig. 5c).

This is a nice addition.

REC8 Trim-Away is an established technique in mouse oocytes and has been reported in multiple studies (Clift et al., Cell 2017, Dunkley et al., Sci Adv. 2023). Furthermore, because we can collect only 5-10 oocytes per *Peromyscus* mouse, confirming the REC8 degradation in oocytes for example by westernblot would require sacrificing ~40 female *Peromyscus* mice, which requires months/years of breeding. Therefore, we believe that the sister-chromatid separation phenotype serves as an indicator for REC8 degradation.

I was thinking antibody staining of oocytes (IF) would be the way to go. Why not do this? Maybe there is a technical limitation I do not notice. For example, the new Figure 5C could have a REC8 stain. Can TRIM-mCherry be seen at all in these images? Clearly Western is not the right experiment, agreed.

Minor changes:

Overall, the language can be simplified/toned down. Just to take the first example: "remarkably" in line 25 – not necessary (since the remark is made, of course), and can be removed with profit. A second early example: "play a pivotal role" in line 42 could be simply "drive" or "direct."

"Tolerant" on line 155 should be "resistant."

Line 213: In the discussion, "direct roles of centromeric satellites" is vague. What is meant here, I think, is simply that we can't easily study non-kinetochore satellite functions in "normal" situations.

Line 227: “(2) what is the evolutionary advantage...” Is this a valid assumption? I would probably just say “Is there an evolutionary advantage, and what is it if so?” I guess we assume dual PMSat chromosomes came from BFB cycles that resolved into this weird persistent arrangement, but is there evidence that they have been preferentially preserved? Would these mice be better off just getting rid of one or the other (a possible experiment mentioned in the text)?

Reviewer #2

(Remarks to the Author)

The authors have appropriately addressed my comments. Congratulations on the beautiful work!

Reviewer #3

(Remarks to the Author)

The original manuscript by Pan et al (Meiosis-specific decoupling of the pericentromere from the kinetochore) has been re-submitted in a revised version. The revised version with the new title (Meiosis-specific distal cohesion site decoupled from the kinetochore) addresses in sound way the remarks and suggestions I had made in my original review. There are only a few points that I feel should be corrected for the final version.

1) The authors have changed the term 'ectopic pericentromere' almost throughout the main text of the paper. However, they should also check the Figure legends. E.g. Figure 6 still carries the misleading caption 'Ectopic pericentromere formation'; line 792. In line 207, I would suggest to write 'ectopic telomere-proximal pericentromere-like structures' instead of just 'pericentromere-like structures' to clearly distinguish between the bona fide pericentromere and the additional telomere-proximal structures. In line 198 I suggest to change the subtitle 'The formation of Additional cohesion site is specific to meiosis' to 'The formation of ectopic additional cohesion sites is specific to meiosis'

2) The authors have removed in the discussion section the sentence describing the reproductive lifespan in different mouse species. While this deletion makes sense, it isolates the new sentence referring to reproductive lifespan at the end of this paragraph (lines 255/256). To better understand the connection between ectopic cohesion and reproductive lifespan I would introduce in line 255 the following sentence (or similar): 'Re-inforced cohesion of smaller chromosomes might help to extend the reproductive lifespan of female mice, as it would prevent deleterious missegregation of chromosomes during the meiotic divisions.'

3) Reviewer#1 rightfully pointed out that Stephen Taylor was misspelled in the Acknowledgement section as 'Stephan Taylor'. Well, the first name is still misspelled and also in the Methods section (line 440) and twice in the 'Reporting Summary' document (Antibodies section).

Reviewer #1 (Remarks to the Author):

This is a review of the manuscript entitled “Meiosis-specific decoupling of the pericentromere from the kinetochore” from Takashi Akera’s laboratory.

The main finding of this paper is that meiotic sister chromatid cohesion is separable from mitotic kinetochore assembly. In addition to this top-level finding, there are other interesting and important other findings. These include: crossover regulation prevents exchange between the two repeat arrays; meiotic sister cohesion depends on PP2A recruitment at the non-kinetochore array; other pericentromere factors are recruited to the non-kinetochore array; BUB1 drives establishment of the telomeric pericentromere. These are amazing findings. They lead to many more questions, which the authors are likely to investigate in future work. These questions are of major importance to multiple chromosome biology fields and include: How are sites of cohesin retention specified in meiosis I? How is the kinetochore assembly site specified during successive mitotic divisions? How is the kinetochore assembly site “remembered” during the meiotic divisions? How do different organisms “know” where to allow crossing over, and how are crossovers “counted?” Though these are old questions, they are important ones, and the current work provides a new experimental system with which study them.

The paper is well-written and clearly reasoned. There are several points on which the data is suggestive but not ironclad (for example, that cohesion “switches” to a new location), so I advise the authors to constrain their claims as much as possible. Even with simplified claims, the manuscript stands out as an important and exciting contribution. Here, I make suggestions primarily related to presentation. I support publication of this manuscript with minor revisions.

Thank you so much for supporting our work! We have constrained our claims across the manuscript. Please see below for our detailed response.

Major questions:

- A main finding is that telomeric PMSats remain cohered in MII due to cohesin retention throughout MI. If the authors were to label another region of the chromosome with dCas9 (histone genes? Other heterochromatin region?) that is not under tension, would this appear to be cohered in MII? Can we be sure the telomeric PMSat is truly cohered and not just “not-under-tension?” Agreed that the separation pattern at the PMSat is different in mitosis and meiosis as shown, but these are different experiments from a technical perspective.
 - o Experiments to address this: REC8 TRIM-away (see comments below) and okadaic acid (also see below)
 - o Perhaps there is not a perfect experiment short of recruiting an inducible Cas9-TEV, but the authors should address this difficulty in the text.

Thanks for pointing this out! As suggested by the reviewer, we attempted to label another region of the chromosome, telomeres, which are present on all chromosomes (including dual PMSat chromosomes) and often heterochromatinized. We tested three approaches to visualize telomeres: an established TALE construct that targets telomere repeats (Miyanari et al., *Nat Struct Mol Biol.* 2013), a PNA probe that specifically recognizes telomere repeats (Markiewicz-Potoczny et al, *Nature* 2021), and a validated antibody against a telomere protein, TRF1 (Markiewicz-Potoczny et al, *Nature* 2021). However, none of these methods robustly labeled telomeres in *Peromyscus* oocytes despite labeling telomeres in *M. musculus* (see **Reviewer’s Fig.1** below). The difference could be partly due to the significantly longer telomere length in *M. musculus* compared to other rodents, allowing robust telomere labeling. Although we were not able to label another locus at the moment, we provide further analyses from the REC8 Trim-Away experiment (Extended Data Fig. 5c,d), which supports our idea that telomeric PMSat is cohered by REC8 cohesin in meiosis II (see below).

Reviewer's Fig. 1 TRF1 staining in *Mus musculus* and *Peromyscus maniculatus* cells. TRF1 antibody recognized telomeres in *M. musculus* cells but not robustly in *Peromyscus* cells, including meiosis II oocytes. Note that TRF1 signals in *P. maniculatus* oocytes are non-specific staining outside chromosomes.

We agree that the ideal experiment would be to have a REC8-TEV transgenic *Peromyscus* mice and induce the expression of Cas9-TEV protease to have spatio-temporal control of cohesin cleavage. As suggested, we addressed the difficulty of this experiment, since robust pipeline to generate transgenic *Peromyscus* mice have not been developed at the moment (Line 270).

- There is a disconnect between the language and the images shown in figure 2A. The authors say the major cohesion site switches over to the telomeric PMSat arrays in the dual PMSat chromosomes in meiosis II, which implies that centromeric REC8, though it persists to MII in these chromosomes, is not as cohesive as it is in normal chromosomes. However, the HEC1 dots in Figure 2b are separated to a similar degree in standard and dual chromosomes. This is confusing. REC8 persistence at the telomeric array is very cool, but it's not entirely clear what the function of either pool truly is in meiosis II.

Thank you for the comment! Based on this comment and a similar comment from Reviewer 3, we have now performed additional quantifications analyzing cohesion at PMSat loci (line scans in Fig. 2a and the sister-kinetochore distance measurement in Extended Data Fig. 3c). These analyses showed that the sister-chromatid cohesion is weaker in oocyte meiosis II compared to mitosis, particularly for dual PMSat chromosomes, consistent with our model.

Images in Fig. 2b are chromosome spreads where chromatid separation are unclear due to the absence of spindle-microtubule pulling forces. We have clarified this point in the figure. Thank you for pointing this out.

Because of the generally weaker cohesion in meiosis II compared to mitosis (Extended Data Fig. 3c), the additional meiosis-specific cohesion site at telomeric PMSat is beneficial to ensure robust sister-chromatid cohesion. It remains unknown why internal PMSat has a slightly, but significantly weaker cohesion compared to standard chromosome centromeres despite they enrich similar PP2A levels (Fig. 3a). We speculate that other cohesion regulators such as I2PP2A/SET, which inhibit PP2A activity could be differentially regulated between internal PMSat and standard chromosome centromeres in oocyte

meiosis. Based on this comment, we have rephrased the sentences to avoid confusions (Line 105-112, 272-276).

Minor points:

- Line 90: “demonstrating the epigenetic memory across soma and germline” Wouldn’t one need to show the telomeric PMSat is competent for CENP-A assembly to know this for sure?

Thank you for pointing this out. We have modified the sentence to avoid confusion (Line 85).

- Line 144: “ectopically” seems to me to be the wrong word, as it implies incorrect or non-natural localization.

We have removed “ectopically” from the sentence (Line 146).

- The TRIM-away experiment for REC8 is nice, but it is not described at which stage the injection was done, and the MI phenotype is not shown. One assumes there are already major problems at this stage.

Thanks for the comment! We microinjected TRIM21-mCherry mRNA and the REC8 antibody at the GV stage of MI (prior to the nuclear envelope breakdown) because of the technical difficulty microinjecting *Peromyscus* MII oocytes at the moment. Based on this comment, we revised the method section (Line 411) and also showed that there is no detectable chromosome separation in MI, likely because the TRIM21-mCherry expression is not high enough at this stage (Extended Data Fig. 5c).

There is also no direct confirmation that REC8 is depleted. This experiment is a nice way of confirming that REC8 holds sisters together in MII (whether at PMSat or centromeres), but it could be tightened up slightly, at least in the way it is described.

REC8 Trim-Away is an established technique in mouse oocytes and has been reported in multiple studies (Clift et al., *Cell* 2017, Dunkley et al., *Sci Adv.* 2023). Furthermore, because we can collect only 5-10 oocytes per *Peromyscus* mouse, confirming the REC8 degradation in oocytes for example by westernblot would require sacrificing ~40 female *Peromyscus* mice, which requires months/years of breeding. Therefore, we believe that the sister-chromatid separation phenotype serves as an indicator for REC8 degradation.

- Image analysis: for figure 2A and similar, I am confused about how the authors distinguished cohered vs. non-cohered PMSat regions. For instance, the PMSat regions on the “standard” chromosome in 2A (MII) look quite separated and are counted as together (not quantified), but then in the “dual” examples, similarly separated, and these are counted as separated. At least the methods section should explain these image-based decisions clearly.

As mentioned above, we now performed additional quantifications to analyze cohesion at PMSat, including standard chromosomes (line scans in Fig. 2a and sister-kinetochore distance measurement in Extended Data Fig. 3c). These analyses showed that cohesion is in general weaker in meiosis II compared to mitosis (two separated PMSat peaks in meiosis II in contrast to a single peak in mitosis (Fig. 2a) and the larger sister-kinetochore distance in meiosis II (Extended Data Fig. 3c)), as pointed out by the reviewer and also consistent with previous studies using standard lab mice (Kim et al., *Nature* 2015, Kouznetsova et al., *EMBO reports* 2019). These analyses also show that cohesion is slightly weaker for internal PMSat of dual PMSat chromosomes compared to standard chromosomes (Extended Data Fig. 3c). We have replaced the previous “cohered vs non-cohered” analysis with the new analyses in Fig. 2a and modified the sentences to clarify these points (Line 105-112). We hope these changes address your concerns.

- Okadaic acid experiment: this is a nice demonstration that phosphatase activity protects sister cohesion that survives to MII. However, without imaging REC8 protein, it is not conclusive that it is cohesin that holds together the telomeric PMSats. The authors should mention this clearly.

We now mention this in the text as part of the Limitations of the Study paragraph (Line 272).

- Figure 5 – the imaging, especially for the 5-Itu experiments, are not totally clear to me. An additional point: can the quantification in 5b and in 5d be placed next to/on top of each other to show off the difference with the two compounds more clearly? This is a key comparison.

Thanks for this point! Now we have both Fig. 5b and 5d (new Fig. 5c and 5e) experiments performed using MI oocytes to be consistent and reorganized the figure panels, so that these figure panels share similar organizations. In this organization, the two graphs are next to each other, as suggested by the Reviewer.

- Can the schematic from extended data figure 4c be moved to main? This is such a nice illustration of what is going on.

Although we greatly appreciate your support to “promote” this schematic to the main figure, we prefer to keep it in the Extended Data Figure because of the lack of molecular markers of recombination as pointed out by Reviewer 3 (see below).

- Acknowledgments: “Stephen Taylor” misspelled as “Stephan”

Done!

- Throughout the manuscript, there are very minor grammatical/punctuation errors that can be corrected in copy editing/proofing.

Done!

Note: I do not have experience in oocyte imaging, so I urge the editors to consult with a reviewer or external advisor with experience in this area to fully evaluate methodological details and the quality of the data. For the sake of clarity: these look good to me but should be checked by a true expert in this area.

Reviewer #2 (Remarks to the Author):

Pan et al. investigate unconventional pericentromere specification separate from the kinetochore in *Peromyscus* mouse oocytes. They show that PMSat, satellite repeats found at centromeres in various *Peromyscus* species, serves as the DNA scaffold for kinetochore assembly by enriching CENP-A. In both mitosis and meiosis, although some chromosomes have two PMSat loci, one internal and one telomeric, the kinetochore is assembled only on the internal PMSat. Interestingly, however, telomeric PMSat acts as the major persistent cohesion site between sister chromatids at meiosis II. Telomeric PMSat enriches the cohesin protector PP2A at meiosis I, together with the pericentromeric proteins Aurora (CPC) and MCAK. The recruitment of CPC and MCAK depends on the BUB1-H2A-pT121 pathway.

Overall, this manuscript describes a novel and interesting observation of pericentromere specification spatially separated from the kinetochore. Although the mechanism by which telomeric PMSat can enrich pericentromeric proteins specifically in meiosis is not yet very clear, they provide a molecular pathway

required for the enrichment. The discussion of the possibility that telomeric PMSat acts as a backup block for smaller chromosomes to protect sister chromatid cohesion against aging is very interesting. The main claims of the manuscripts are well supported by appropriately designed experiments. The results presented in the manuscript will attract a wide range of readers in the field of chromosome biology and reproductive biology. I would support publication in Nature Communications if the authors could address the following comments.

We appreciate your constructive feedback and your perspective on the broader impact of our work! Please see our detailed responses to your comments.

1. The authors claim that telomeric PMSat is the major persistent cohesion site at meiosis II. In support of this, they show that PMSat cohesion is more tolerant to treatment with the PP2A inhibitor Okadaic acid (OA). However, the effect of OA treatment is not specific to cohesion protection and is unlikely to mimic physiological conditions. I wondered if they could test REC8 TRIM-Away to partially degrade cohesin, which could mimic a condition of aged oocytes, and see if the PMSat site helps maintain sister chromatid cohesion. The authors have successfully achieved REC8 TRIM-Away (Fig. 2c), so it seems possible to find a condition where REC8 is partially degraded. If such an experiment is technically difficult, it could be informative to show how unseparated sister chromatids looked like after REC8 TRIM-Away in Fig. 2c - did they have persistent cohesion at telomeric PMSat?

Thank you for the comment! Because our REC8 Trim-Away (Fig. 2c) is a partial REC8 degradation (~30% of sister chromatids remained cohered), we have analyzed how these chromatids are connected, focusing on dual PMSat chromosomes (Extended Data Fig. 5d). The analysis showed that unseparated dual PMSat chromosomes are predominantly connected at telomeric PMSat, supporting our model.

2. Related to the above, can they show quantification of REC8 signals on internal and telomeric PMSat, at meiosis I and II?

Thanks for the suggestion! We have now added line scans of REC8 signal intensities along the meiosis I chromosome (Extended Data Fig. 5a) and a quantification of REC8 localization pattern in meiosis II (Fig. 2b). Line scans in MI showed a slight enrichment of REC8 cohesin around the PMSat locus except for the kinetochore region. Quantification in meiosis II showed that the majority of dual PMSat chromosomes has REC8 localization at both internal and telomeric PMSat, supporting our idea.

Minor points:

1. In figure panels, it is not clear what area of the whole image is magnified in the inset images. The authors could consider putting a rectangle to indicate the magnified area. It would be informative to include a scale bar for the magnified images.

Thank you for pointing this out. Based on this suggestion, we have added rectangles to indicate the magnified area in each image.

2. Fig. 6a-c. If I understand correctly, these line scan data show single representative chromosomes, although according to the legend the authors have images of 13, 21, 11 cells from three independent experiments. Consider adding a quantification of fluorescence signals measured on internal and telomeric PMSats in all the images, to be consistent with what is done in Fig. 4.

Done!

Reviewer #3 (Remarks to the Author):

This manuscript by Pan et al (Meiosis-specific decoupling of the pericentromere from the kinetochore) describes a functional characterization of chromosomes that harbor in addition to the regular pericentromeric regions, similar regions elsewhere in the same chromosomes, close to the telomeres. These chromosomes were analyzed in mouse species of the genus *Peromyscus*, approximately separated by 25 million years of evolution from the common laboratory mouse *Mus musculus*. It has been known for quite some time that the low-complexity DNA sequence underlying the centromeric regions in many higher eukaryotes is neither sufficient nor absolutely required for establishment of a functional centromere. This has led to the notion that centromeres are determined epigenetically, mainly defined by the presence of the centromeric H3 variant CENP-A. CENP-A deposition results in the recruitment of other proteins, which are part of the inner kinetochore and which in turn serve as an assembly platform for the KMN complex that mediates attachment of microtubules of the spindle apparatus during cell divisions. The extended regions surrounding the core centromere – the pericentromeres – are also characterized by repetitive regions, retroelements, transposons and a low density of expressed genes in these heterochromatic settings. For a functional aspect, pericentromeres are known to be enriched in factors providing sister chromatid cohesion – the cohesin complex. The sequences underlying the pericentromeric regions are also not exclusively found in this area, but information about functional relevance of these sequences elsewhere in the genome is scarce. Here, the authors provide evidence that these regions do not suffice to assemble kinetochores but are enriched in factors required for establishment and maintenance of sister chromatid cohesion. Most interestingly, this does not apply for mitotic divisions, but appear to be specific for meiotic divisions (female meiosis it is; male meiosis was not looked at).

The studies were mainly conducted using the mouse species, *Peromyscus maniculatus*. The typical pericentromeric sequence signature (called PMSat), which usually occurs only once around the centromere, was found twice in three out of 25 chromosomes (called dual PMSat chromosomes) in this species. The authors diligently used various approaches to ascertain the presence of bona fide PMSat sequences at regions distal to centromeres (in the following telomere proximal PMSat). These telomere proximal PMSats were thoroughly confirmed by either FISH experiments or by targeting a nuclease dead version of Cas9, fused to EGFP, to PMSat regions. Combined FISH/IF and Autofluorescence/IF experiments revealed that the telomere proximal PMSat sequences do not trigger the formation of kinetochores as evidenced by a lack of staining against the outer kinetochore marker HEC1. However, these regions do mediate cohesion up to the metaphase-anaphase transition, but do this exclusively in female meiosis. This is exemplified by visualizing individual metaphase II chromosomes, which are cohesed at the telomere proximal PMSat but not at the centromere. These type of chromosomes were not observed during mitotic divisions. This cohesion is dependent on cohesin as shown by TRIMaway mediated destruction of the cohesin kleisin subunit REC8 resulting in cohesion loss. The persistence of cohesion at these sites up to metaphase of meiosis II requires cohesin protection from the onslaught of separase during meiosis I. The authors show that protection is due to recruitment of PP2A to telomere proximal PMSat. This recruitment is paralleled by localization of components of the CPC and of the kinesin-13 MCAK to the telomere proximal PMSats. The authors also provide evidence that this ectopic recruitment only occurs in animals carrying PMSats, as it is also observed in chromosomes derived from *P. polionotus*, which also carries three chromosomes with telomere proximal PMSats, but not in chromosomes from *P. californicus*, which does not contain chromosomes with extra regions containing PMSats. To gain insight into factors required for the ectopic localization of typical pericentromeric factors at the ectopic PMSats, the authors inhibited two kinases known to introduce histone phosphorylations involved in recruitment of CPC and MCAK. Notably, inhibition of BUB1, which phosphorylates histone H2A at threonine 121, significantly reduced the CPC component, and (less convincing – see below) MCAK at the telomere proximal PMSats.

I feel that the manuscript thoroughly addresses the relevant questions, the experiments are carried out in a sound way and the conclusions drawn are of significant interest for a broad readership. However, I do have a few issues that need to be addressed before the paper can be published.

Thank you so much for providing constructive feedback! We have addressed all the comments, which significantly improved the manuscript.

1. Pericentromeres are defined as the genomic regions surrounding the centromere (from the Greek word peri meaning around, surrounding). Thus, the occurrence of DNA sequences underlying pericentromeres elsewhere in the genome, far away from centromeres, does not justify calling these regions “pericentromeres”, even if they resemble bona fide pericentromeres in a functional sense. Additional ectopic and functional centromeres are called neocentromeres. Thus, I would strongly suggest that the authors do not call the telomere proximal PMSat containing regions pericentromeres as they do at numerous occasions in the text. Even the title of the paper is misleading in this respect. I would suggest sth like “Meiosis-specific persistence of chromatid cohesion at distal regions containing pericentromere-specific sequence repeats” I acknowledge that this sounds somewhat bulky, but it is less misleading. Also “pericentromere specification” (found e.g. in the abstract, lines 18 and 32) and “rewiring pericentromeres” (lines 180 and 193) should be replaced by an alternative term. This is particularly important, since the present studies have looked mainly at one functional aspect of pericentromere function – cohesion. But there are other functions the authors have not investigated. For example, pericentromeric heterochromatin plays a role in the three dimensional packaging of chromatin within the nucleus – chromocenter formation (e.g. Jagannathan et al (2018) <https://doi.org/10.7554/eLife.34122>)

Thanks for this suggestion. We have now changed our title, abstract, and the main text not to call telomeric PMSat pericentromeres. We have changed them to either “additional cohesion site” or “pericentromere-like structure” depending on the context of each sentence.

2. Along those lines... “Colocalization” in IF experiments describes the fact that signals corresponding to the same pixels in two channels of the same R.O.I.s indicate presence of two proteins at the same spot within the cell. The CPC is known to localize to the inner centromere – in fact, the CPC component INCENP has derived its name exactly because of this localization pattern. This is exactly, what the authors show, e.g. in Extended Fig. 8. The signals for HEC1 and MCAK/Survivin/pAurora are distinct. Thus, it is not correct to call the localization pattern of CPC components as “colocalization with the kinetochore” (lines 201, 205 and 703).

Thank you so much for catching this! Now, we have modified the sentences, describing that CPC localizes between sister kinetochores.

3. The specificity of the ectopic cohesion site in female meiosis is intriguing. However, the authors need to point out that they can make this point only for female meiosis, as they have not looked at male meiosis

Thanks for pointing this out. We clarified that our findings are in female meiosis in the main text (Line 113 and 199).

4. In the introduction (lines 41 ff) the authors dwell on cohesion as being restricted to the centromeric/pericentromeric regions. However, this is only true for cohesion at metaphase; cohesion is established along the entire chromosomes and removed in steps (prophase pathway vs. proteolytic degradation at the metaphase-anaphase transition). The authors fail to mention this.

Based on this comment, we now describe the prophase pathway in the Introduction and the Result section (Line 39-41, 127-129).

5. Line 381. The terms “left” and “right” do not correspond to the assembly in Figure 6. Line 382. Asterisks are not present in Fig. 6.

Done!

6. Figure 4. The schematic would be much more appropriate to be part of Figure 5, as the BUB1-H2ApT121 and Haspin-H3pT3 pathways are investigated in Figure 5.

Done!

7. Some of the critical data shown in Figure 5 are not represented appropriately in the text or are not convincing. The H2A-pT121 staining in Fig 5c shows a strong accumulation of the signals at the pericentromeres. This is not mentioned in the text (“signals were detected along the chromosome” – line 190). In addition, the reduction of the MCAK signals on dual chromosomes after BUB1 inhibition does not look too convincing, particularly because in the control case (Fig. 5e, upper panel), a distinction between internal and telomeric MCAK signals is almost impossible. I would suggest providing a more appropriate image.

Thanks for pointing out! Now we mention about the H2A-pT121 accumulation at pericentromeres (Line 196) and provide more representative images for Fig. 5e (new Fig. 5f).

8. The suggestion that the presence of the telomere proximal PMsats may result in a more stable cohesion of small chromosomes during female meiosis (metaphase arrest) is intriguing. The authors support this notion by a significantly extended reproductive lifespan of ectopic PMsats containing *P. maniculatus* and *P. polionotus* individuals when compared to *M. musculus*. This may be, however, a pure coincidence, particularly given the evolutionary distance of 25 million years. It would be more informative to look at the reproductive lifespan of the closely related species *Peromyscus californicus*, which also lacks ectopic PMsats. In addition, even if it is technically very challenging, one could at least mention an experimentally based answer to this question: Introduce ectopic PMsats in the *P. californicus* genome or remove/inactivate these regions in *P. maniculatus* and assess whether this has an influence on the reproductive lifespan.

Thank you so much for this comment! We now mention in the Discussion section how we could experimentally test the significance of telomeric PMsat on reproductive lifespan (Line 255). Regarding the reproductive lifespan of *P. californicus*, we could not find any evidence in the literature that their reproductive lifespan is shorter than *P. polionotus* or *P. maniculatus*, and therefore, we removed this sentence from the Discussion.

9. The quantification of PP2A signal intensities on control (standard) chromosomes is identical to the internal pericentromeres on dual chromosomes (Fig. 3A). This should result in similar protection of cohesion against separase activity in meiosis I. How do the authors then explain the frequent occurrence of dual chromosomes, which are separated at the centromeric regions in meiosis II (Fig. 2A)?

We agree with the reviewer that centromeres of standard chromosomes and internal PMsat of dual PMsat chromosomes enrich similar PP2A levels (Fig. 3a). We performed additional analyses to examine sister-chromatid cohesion, which confirms that sister kinetochores are slightly more separated for internal PMsat compared to standard chromosomes in meiosis II despite they enrich similar PP2A levels (Extended Data Fig. 3c, also see the responses to Reviewer 1’s comments). The underlying mechanism remains unknown, but one possibility is that other cohesion regulators such as I2PP2A/SET, which inhibits PP2A activity, could be differentially regulated between internal PMsat and standard chromosome centromeres specifically in oocyte meiosis. We now made these points clearer in the revised manuscript (Line 272) and also modified the chromosome diagrams in the figures to avoid confusion.

It is intriguing that centromeric cohesion is in general weaker in meiosis II compared to mitosis (Fig. 2a and Extended Data Fig. 3c). This weaker cohesion could be a driving force to create additional cohesion sites at telomeric PMSat to ensure sister-chromatid cohesion. We have added this discussion to the Discussion section (Line 247). We appreciate this comment!

10. Extended Fig. 2b. The CENP-A signals of the selected dual chromosome are almost invisible. The assignment then of internal vs. telomeric PMSat appears somewhat arbitrary. If the very weak CENP-A signals are used for this distinction, one could also argue that in this case all PMSat regions exhibit HEC1 signals (4 distinct dots in the anti-HEC1 staining).

CENP-A signals become significantly dimmer when combined with the FISH technique. We replaced this figure with an immunofluorescence of CENP-A and HEC1 without PMSat FISH to show that CENP-A and HEC1 always colocalize on the chromosome (new Extended Data Fig. 2b). Together with the observation that HEC1 consistently localizes at internal PMSat in mitosis (Fig. 1c), these results collectively suggest that CENP-A localizes at internal PMSat.

11. Extended Fig. 4c. It is an important point that recombination between the centromere and the telomere proximal PMSat is deleterious. The authors have looked at a total of 18 cells, meaning 54 potential candidate chromosomes and they have seen no indication for recombination based on IF images. Given the small size of these chromosomes, it would be hard to detect such a constellation anyways. It would be much more informative to look at molecular markers and determine recombination rates.

I agree with the reviewer that it would be informative to label recombination markers to carefully analyze the recombination pattern. However, such experiments require establishing a new experimental system to look at early prophase I (i.e., embryonic ovary for female meiosis and testis for male meiosis). Therefore, we believe it is beyond the scope of the current manuscript.

Reviewer #1 (Remarks to the Author):

Pan et al. have address most of my comments, including major concerns.

I have the following responses to comments and make minor suggestions for improving the manuscript.

Based on this comment, we revised the method section (Line 411) and also showed that there is no detectable chromosome separation in MI, likely because the TRIM21-mCherry expression is not high enough at this stage (Extended Data Fig. 5c).

This is a nice addition.

REC8 Trim-Away is an established technique in mouse oocytes and has been reported in multiple studies (Clift et al., Cell 2017, Dunkley et al., Sci Adv. 2023). Furthermore, because we can collect only 5-10 oocytes per Peromyscus mouse, confirming the REC8 degradation in oocytes for example by westernblot would require sacrificing ~40 female Peromyscus mice, which requires months/years of breeding. Therefore, we believe that the sister-chromatid separation phenotype serves as an indicator for REC8 degradation.

I was thinking antibody staining of oocytes (IF) would be the way to go. Why not do this? Maybe there is a technical limitation I do not notice. For example, the new Figure 5C could have a REC8 stain. Can TRIM-mCherry be seen at all in these images? Clearly Western is not the right experiment, agreed.

Thank you for this comment to show REC8 IF after REC8 TrimAway. Since REC8 TrimAway is an established method in mouse oocytes, we did not perform REC8 IF to minimize the use of the REC8 antibody, which is a custom antibody with limited supply. We routinely confirm TRIM21-mCherry signals prior to the fixation to assess the success rate of microinjection. However, the signals become significantly weaker after the fixation and therefore we cannot include them to the current figures.

Minor changes:

Overall, the language can be simplified/toned down. Just to take the first example: “remarkably” in line 25 – not necessary (since the remark is made, of course), and can be removed with profit. A second early example: “play a pivotal role” in line 42 could be simply “drive” or “direct.”
We have toned down the language as advised.

“Tolerant” on line 155 should be “resistant.”

Done!

Line 213: In the discussion, “direct roles of centromeric satellites” is vague. What is meant here, I think, is simply that we can’t easily study non-kinetochore satellite functions in “normal” situations.

Done!

Line 227: “(2) what is the evolutionary advantage...” Is this a valid assumption? I would probably just say “Is there an evolutionary advantage, and what is it if so?” I guess we assume dual PMSat chromosomes came from BFB cycles that resolved into this weird persistent arrangement, but is there evidence that they have been preferentially preserved? Would these mice be better off just getting rid of one or the other (a possible experiment mentioned in the text)?

Yes, it remains unclear if dual PMSat is beneficial or not, so we have rephrased the sentence accordingly.

Reviewer #2 (Remarks to the Author):

The authors have appropriately addressed my comments. Congratulations on the beautiful work!

Thanks a lot!

Reviewer #3 (Remarks to the Author):

The original manuscript by Pan et al (Meiosis-specific decoupling of the pericentromere from the kinetochore) has been re-submitted in a revised version. The revised version with the new title (Meiosis-specific distal cohesion site decoupled from the kinetochore) addresses in sound way the remarks and suggestions I had made in my original review. There are only a few points that I feel should be corrected for the final version.

1) The authors have changed the term 'ectopic pericentromere' almost throughout the main text of the paper. However, they should also check the Figure legends. E.g. Figure 6 still carries the misleading caption 'Ectopic pericentromere formation'; line 792. In line 207, I would suggest to write 'ectopic telomere-proximal pericentromere-like structures' instead of just 'pericentromere-like structures' to clearly distinguish between the bona fide pericentromere and the additional telomere-proximal structures. In line 198 I suggest to change the subtitle 'The formation of Additional cohesion site is specific to meiosis' to 'The formation of ectopic additional cohesion sites is specific to meiosis'

Done!

2) The authors have removed in the discussion section the sentence describing the reproductive lifespan in different mouse species. While this deletion makes sense, it isolates the new sentence referring to reproductive lifespan at the end of this paragraph (lines 255/256). To better understand

the connection between ectopic cohesion and reproductive lifespan I would introduce in line 255 the following sentence (or similar): 'Re-inforced cohesion of smaller chromosomes might help to extend the reproductive lifespan of female mice, as it would prevent deleterious missegregation of chromosomes during the meiotic divisions.'

Done! Thanks for the suggestion.

3) Reviewer#1 rightfully pointed out that Stephen Taylor was misspelled in the Acknowledgement section as 'Stephan Taylor'. Well, the first name is still misspelled and also in the Methods section (line 440) and twice in the 'Reporting Summary' document (Antibodies section).

I apologize for these misspelling. We have fixed the issues.